# Energy requirements and carbon emissions for a low-carbon energy transition

**Aljoša Slameršak** [1]✉**, Giorgos Kallis** [1,2] **& Daniel W. O'Neill** [3]

Achieving the Paris Agreement will require massive deployment of low-carbon energy. However, constructing, operating, and maintaining a low-carbon energy system will itself require energy, with much of it derived from fossil fuels. This raises the concern that the transition may consume much of the energy available to society, and be a source of considerable emissions. Here we calculate the energy requirements and emissions associated with the global energy system in fourteen mitigation pathways compatible with 1.5 °C of warming. We find that the initial push for a transition is likely to cause a 10–34% decline in net energy available to society. Moreover, we find that the carbon emissions associated with the transition to a low-carbon energy system are substantial, ranging from 70 to 395 $GtCO_2$ (with a cross-scenario average of 195 $GtCO_2$). The share of carbon emissions for the energy system will increase from 10% today to 27% in 2050, and in some cases may take up all remaining emissions available to society under 1.5 °C pathways.

The IPCC's Special Report on Global Warming of 1.5 °C concludes that we can still meet the 1.5 °C target and that by doing so, we would reduce climate impacts and limit the risk of exceeding the tipping points of the climate system[1]. The report provides a range of low-carbon energy pathways compatible with limiting global warming to 1.5 °C. However, at present, there is no estimate of how much energy would be needed to build and maintain a low-carbon energy system, or what amount of greenhouse gas emissions would be associated with such a transition[2–4]. This is an important gap in knowledge, as previous research suggests that rapid growth of low-carbon infrastructure could use a substantial amount of the global energy supply[5,6]. Moreover, since the global energy supply is currently derived mostly from fossil fuels, the transition itself may become a source of significant emissions[7,8].

Some studies suggest that renewables have a lower energy return on energy invested (EROI) compared to the current energy system[9,10]. Lower EROI implies less energy delivered to society relative to the energy required to supply the energy, leading these studies to conclude that a low-carbon energy transition may result in less energy available to society. The energy required for the transition might push society into an "energy–emissions trap", where achieving ambitious

climate mitigation could lead to a period of reduced energy availability[11,12], and at the same time, also consume a large share of the remaining carbon budget[13]. Recent studies, however, find the hypothesis of lower energy availability might be exaggerated due to overestimating the EROI of fossil fuels[14,15] and underestimating improvements in the EROI of renewable energy technologies[16,17].

Alongside EROI, life-cycle assessment is another accounting technique that has been used to quantify climate change impacts from different energy generation technologies. However, life-cycle studies typically only estimate the impacts of present-day energy technologies applied to a particular case study[18–20]. Life-cycle assessment has rarely been used in a dynamic analysis where the impacts of technologies change over time, or to assess the cumulative impacts of decarbonising the entire global energy system.

A notable exception is a study by Pehl et al.[21] who used a dynamic approach to estimate the energy requirements and emissions for the construction, operation, and maintenance of power plants. The authors combined a dynamic life-cycle assessment framework with an Integrated Assessment Model (IAM), estimating that emissions associated with power plants would lead to 82 $GtCO_{2eq}$ of cumulative emissions from 2010 to 2050. In another study, Di Felice et al.[8]

[1]The Institute of Environmental Science and Technology, ICTA-UAB, Autonomous University of Barcelona, Barcelona, Spain. [2]Catalan Institution for Research and Advanced Studies, ICREA, Barcelona, Spain. [3]Sustainability Research Institute, School of Earth and Environment, University of Leeds, Leeds, UK. ✉e-mail: aljosa.slamersak@gmail.com

conducted a life-cycle assessment of the indirect emissions associated with the EU's renewable energy strategy, calculating that 25 GtCO$_{2eq}$ would be emitted in the decarbonisation of the EU's electricity generation from 2020 to 2050. These studies, however, only cover electricity generation, which currently represents just ~20% of global final energy use. Moreover, each study only analysed one specific low-carbon pathway.

Here, we estimate how much energy would be required, and how much carbon would likely be emitted, to construct, operate, and maintain the global energy system during a low-carbon energy transition. Our study separates the energy and emissions associated with the energy system from the energy and emissions remaining for other societal uses. We thus provide complementary information to existing mitigation pathways. Moreover, by modelling dynamic changes in the EROI of the energy system in fourteen different mitigation pathways produced by six IAMs, we provide a holistic picture for a range of distinct energy transitions, all in line with the ambitious goal of stabilising global warming below 1.5 °C. We also assess the energy–emissions trap hypothesis, considering the latest literature on the EROIs of different energy technologies. In doing so, we follow a consumption-based accounting approach using an EROI analysis to estimate both direct (on-site) and indirect (upstream) energy use and emissions associated with constructing, operating, and maintaining the energy system and the energy supply to society. Based on our results, we suggest that the energy requirements and emissions of the energy system should be explicitly modelled in the next generation of low-carbon mitigation pathways.

## Results

### Estimating energy requirements and emissions

We refer to the energy that would be required during a low-carbon energy transition as the "energy for the energy system" and the carbon that would be emitted as the "energy system emissions". Energy for the energy system includes the energy required for the construction (including decommissioning), operation, and maintenance of energy facilities like power plants, mines, and refineries, as well as the energy required to transport the energy carriers from the point of extraction to the end-user.

To estimate the energy for the energy system, we apply the method of net energy analysis, calculating energy return on energy invested (EROI) at the final energy stage. EROI at the final energy stage tells us how much of the total final energy is used by the energy system to extract, process, convert, and deliver a unit of energy to the point of use for society[22–24]. Net energy at the final energy stage is defined as the difference between total final energy and the energy for the energy system and represents the part of energy production that can be used for societal work[6]. We calculate EROIs and the energy for the energy system for twenty-seven energy conversion technologies, which cover the entire energy system, from 2020 to 2100 (see Methods). We distinguish between four different energy carriers (electricity, gases, liquid fuels, and solids), following the approach of Arvesen et al.[25]. To obtain the share of energy for the energy system, we divide the energy requirements of the energy system by total final energy.

In our calculations, we combine a range of EROI estimates of present-day energy technologies (see Supplementary Figs. 1 and 2 and Supplementary Table 1), with projections of future changes in EROI due to technological improvements that we estimate using energetic experience curves[16]. To account for the range of present-day estimates, the uncertainty of technological change, and resource availability, we report estimates of the energy requirements for each energy technology using low, median, and high-EROI values representing the first, second, and third quartiles of the inter-quartile range of our estimates, respectively (see Methods). There is a divergence in EROI values in different studies, which can be traced to the distinct definitions of energy system boundaries, which can vary depending on the

research objectives pursued[26]. As a result, EROI values are often not directly comparable between different energy carriers and between different studies[27]. To address this shortcoming, we apply a consistent energy system boundary to all energy technologies. This boundary extends from the point of extraction (primary energy stage) to the point of use (final energy stage), as suggested by EROI analysts[14,28].

To represent a range of plausible EROI transitions, we develop three EROI scenarios. In the high-EROI scenario, we assume a fast increase in the EROI of renewables from the high end of present-day EROI values. In this scenario, we assume the EROI of bioenergy at the primary energy stage remains near the median of present-day values. This scenario can be interpreted as a future of high innovation and broad policy support for renewables, alongside efficient and sustainable harvest of biomass for energy. In the low-EROI scenario, we assume a gradual increase in the EROI of renewables from the median of present-day EROI values, with the EROI of bioenergy remaining near the lower end of present-day values. Such a scenario corresponds to a future of moderate innovation and balanced policy support for renewables, and low efficiency in the management of bioenergy. In the median-EROI scenario, we assume a gradual increase in the EROI of renewables from the median of present-day EROI values, with bioenergy remaining near the median of present-day values. In all three scenarios, we assume a gradual decline in the EROI of fossil fuels at the primary energy stage from the present-day median value towards the present-day low-EROI value, in line with historical trends and the existing literature[5,14,29]. For a detailed overview of the assumptions across all of the EROI scenarios, see Supplementary Tables 2–7 and Supplementary Figs. 3 and 4.

We calculate energy system emissions for different 1.5 °C–compatible mitigation pathways as the product of energy for the energy system and the carbon intensity of the energy system. We divide the energy for the energy system into four energy carriers that each have different carbon intensities and distinguish between three life-cycle stages: construction, operation and maintenance, and decommissioning at the end of lifetime. The carbon intensity of energy carriers changes over time, primarily depending on the share of conventional fossil fuels (i.e. fossil fuel technologies without carbon capture and storage) in the energy mix of each carrier. By combining the effects of technological improvements in the EROIs of energy technologies with changes in carbon intensity due to the declining share of fossil fuels, we capture the dynamic evolution of carbon emissions associated with the energy system over time.

We illustrate our findings using the four illustrative pathways from the IPCC's Special Report on Global Warming of 1.5 °C. Three of these pathways were taken from the Shared Socioeconomic Pathways (SSP-1.9) scenario study[30], while one originates from the Low Energy Demand (LED) scenario[31]. These pathways represent the archetypes of different possible futures in terms of energy use, greenhouse gas emissions, and preferences for energy conversion technologies, yet all manage to stabilise global warming below 1.5 °C (see Table 1). LED and S1-A are pathways of rapid decarbonisation, achieved by phasing out more than 50% of fossil fuel energy by 2040, accelerating growth in renewable energy, and decreasing energy demand. S5-R is a pathway of slower decarbonisation, long-term growth in final energy, and large-scale carbon removal (which compensates for the higher emissions at the beginning of the transition). S2-M is a "middle of the road" pathway that combines decarbonisation with slow growth in final energy and moderate carbon removal. For a complete representation of different 1.5 °C–compatible futures, we also analyse ten additional pathways produced by Rogelj et al.[30] (Supplementary Table 8), and present average values for all fourteen pathways.

Each pathway has different "total cumulative emissions", which depend on the quantity of carbon sequestration it includes[32]. From the perspective that interests us here, the total cumulative emissions that are compatible with 1.5 °C of warming can be partitioned into energy

**Table 1 | The four IPCC illustrative pathways**

| Pathway | Scenario assumptions | Energy mix and emissions |
|---|---|---|
| LED: Low Energy Demand[1,31] | Moderate population growth. Moderate decrease in energy and material use. High innovation and fast adoption of sustainable technologies. Convergence to sustainable, low-carbon diets. | Average annual emissions reduction rate (2020–2040): 6.5% (rapid decarbonisation)<br>Change in energy use (2020–2100): –44%<br>Cumulative negative emissions from BECCS (2020–2100): 0 $GtCO_2$<br>Share of cumulative final energy (2020–2100):<br>—Renewables: 42.8%<br>—Nuclear: 6.9%<br>—Fossil fuels: 37.3%<br>—Bioenergy: 12.9% |
| S1-A: Sustainable Development[30,58,76] | Low population growth. Stable energy consumption and slow material growth. High innovation and fast adoption of sustainable technologies that improve energy efficiency. Convergence to low-waste and low animal share diets. | Average annual emissions reduction rate (2020–2040): 5.5% (rapid decarbonisation)<br>Change in energy use (2020–2100): –7%<br>Cumulative negative emissions from BECCS (2020–2100): 150 $GtCO_2$<br>Share of cumulative final energy (2020–2100):<br>—Renewables: 44.1%<br>—Nuclear: 5.6%<br>—Fossil fuels: 39.9%<br>—Bioenergy: 10.5% |
| S2-M: Middle of the Road[30,59,76] | Moderate population growth. Moderate growth in energy and material use. Gradual institutional and behavioural changes with slower technological innovation. Continuation of historical dietary transition trends. | Average annual emissions reduction rate (2020–2040): 5.0% (moderate decarbonisation)<br>Change in energy use (2020–2100): +40%<br>Cumulative negative emissions from BECCS (2020–2100): 415 $GtCO_2$<br>Share of cumulative final energy (2020–2100):<br>—Renewables: 33.7%<br>—Nuclear: 13.1%<br>—Fossil fuels: 36.2%<br>—Bioenergy: 16.9% |
| S5-R: Fossil-fuelled Development[30,60,76] | Low population growth. High growth in energy and resource use. Delayed energy transition allowed by high innovation and large-scale adoption of negative emissions technologies. Diets with high animal shares and high waste. | Average annual emissions reduction rate (2020–2040): 3.8% (slower decarbonisation)<br>Change in energy use (2020–2100): +76%<br>Cumulative negative emissions from BECCS (2020–2100): 1190 $GtCO_2$<br>Share of cumulative final energy (2020–2100):<br>—Renewables: 37.6%<br>—Nuclear: 8.4%<br>—Fossil fuels: 31.3%<br>—Bioenergy: 22.8% |

The table summarises the fundamental assumptions and characteristics of the four alternative energy transitions that were selected as illustrative pathways in the IPCC's Special Report on Global Warming of 1.5 °C. The aim of the illustrative pathways is to show different possible futures that lead to a stabilisation of global warming. The pathways differ with regards to socioeconomic, behavioural, and technological assumptions. For the "SN-X" pathways, the number N refers to the scenario narrative from the SSPs, while the letter X denotes the model that produced a particular mitigation pathway.
*BECCS* bioenergy with carbon capture and storage.

system emissions and emissions for other societal uses. To obtain the share of energy system emissions in any given year, we divide energy system emissions by total emissions in that year (where the latter is obtained from the pathway data).

**Energy system emissions during the transition are substantial**
We find that the cumulative carbon emissions associated with the energy system during the transition are substantial, and represent a considerable share of total cumulative emissions under different 1.5 °C–compatible scenarios (Fig. 1). The fourteen-pathway average is 195 $GtCO_2$ for the median-EROI scenario and ranges from 185 $GtCO_2$ for the high-EROI scenario to 290 $GtCO_2$ for the low-EROI scenario. These results correspond to an average of 21% of total emissions for the fourteen energy pathways under median-EROI assumptions, or 20% for high- and 31% for low-EROI assumptions.

Figure 1 shows the difference in cumulative energy system emissions among the IPCC's four illustrative pathways. Cumulative emissions for median-EROI values range from 70 $GtCO_2$ (12% of total cumulative emissions) for LED, which is a low-energy-demand/no-BECCS pathway, to 220 $GtCO_2$ (20% of total cumulative emissions) for S5-R, which is a high-energy/high-BECCS pathway. Generally, in slower decarbonisation pathways with higher energy use and higher deployment of BECCS, more carbon emissions are associated with the energy

system during a low-carbon energy transition (see also Fig. 3 and Supplementary Fig. 5).

Energy system emissions become more important over time, as they take up an increasing share of total emissions, leaving fewer emissions for other uses in society. We estimate that the share of energy system emissions will increase to 2–5 times its current value by 2060, depending on different EROI assumptions (Fig. 2). After 2060, the share of emissions stabilises in most pathways, as the pathways achieve a high degree of decarbonised energy. The share of emissions for the energy system in pathways S1-A, S2-M, and S5-R is much higher than in the LED pathway, which completely decarbonises its energy system. The fourteen-pathway average of energy system emissions increases from 10% of total emissions in 2006–2015 (for the median-EROI scenario) to 27% in 2050, and reaches 40% by the end of the century. For the low-EROI scenario, the share increases from around 12% in 2006–2015, to 39% in 2050, and 59% by the end of the century. In the high-EROI scenario, the change is from 9% in 2006–2015 to 26% by 2050, and 31% by the end of the century.

The increase in the share of energy system emissions means that the decarbonisation of the energy system and energy supply is slower than the decarbonisation of the overall economy. A high share of emissions for the energy system may impose—particularly under the low-EROI scenario—a tight constraint on the "residual emissions"

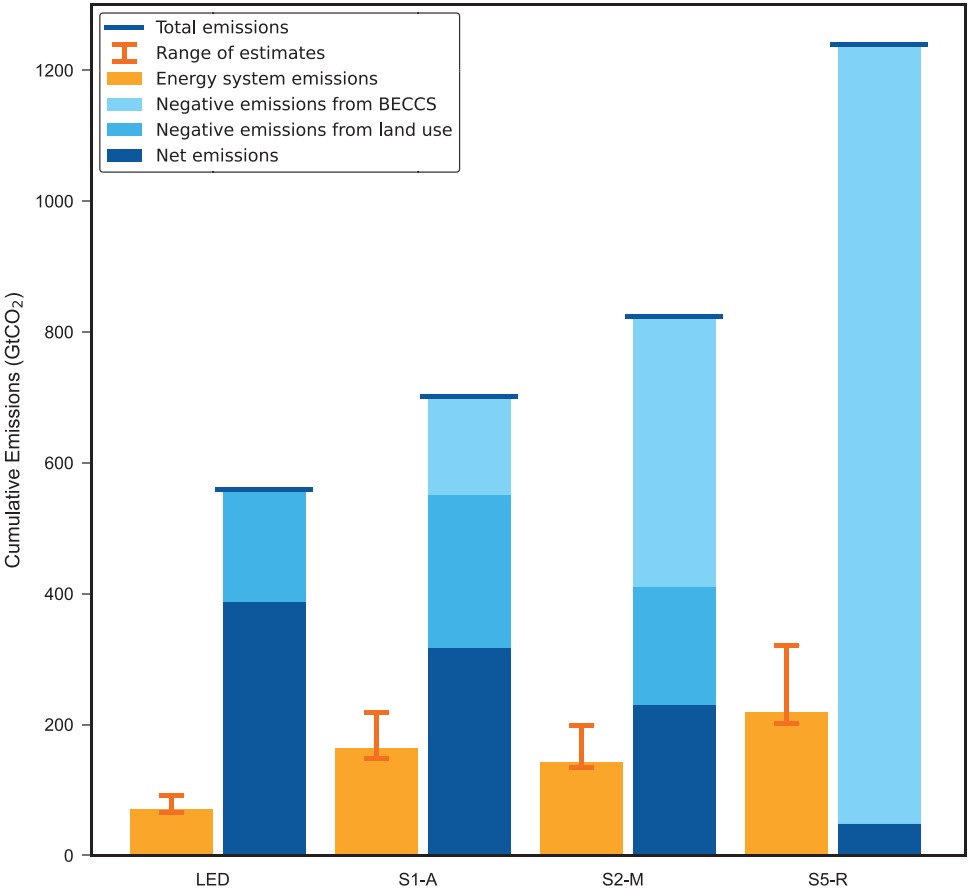

**Fig. 1 | Energy system emissions for each of the four 1.5 °C illustrative pathways.** Energy system emissions (orange columns) are compared to total cumulative emissions (blue columns). Orange error bars indicate the range of energy system emissions calculations from high- to low-EROI model runs. Net emissions are equal to total emissions vented into the atmosphere minus carbon sequestration from BECCS and the land-use sector (AFOLU). Each pathway allows for different total carbon emissions (and hence different total cumulative emissions) as each pathway assumes different amounts of carbon sequestration and non-$CO_2$ greenhouse gas emissions. See Supplementary Fig. 5 for all fourteen 1.5 °C pathways.

remaining for activities such as aviation, steel and cement production, and load-following electricity, which are difficult to decarbonise and currently generate ~9 $GtCO_2$ per year[33,34]. For some of the fourteen pathways, the energy system requires all of the residual emissions by 2080 under all three EROI scenarios—leaving no emissions for activities such as air travel, or steel and cement production.

A high share of energy system emissions in some of the pathways suggests the models may have been overly optimistic in their calculations of residual emissions. If this is the case, then the models need to either reduce the emissions allocated to other economic activities in society or adjust their choice of energy technologies to reduce energy system emissions, as in the LED pathway. The pathways may be defended by assuming that technological innovation will make it possible to cut emissions to zero in the sectors that are difficult to decarbonise today. However, this assumption is highly speculative, given the essential role of fossil fuels in the production of steel and cement, which are critical materials in the economy[25,35].

**Energy transition leads to a small jump in emissions**
Our results suggest that the upfront energy required to build a low-carbon energy system would only lead to a small jump in annual energy system emissions, with the most notable increase taking place in the pathways of higher energy use and continued reliance on fossil fuels beyond 2030 (e.g. S5-R; Fig. 3). Average energy system emissions in the S5-R pathway from 2020 to 2030 are 4.0 $GtCO_2$ per year for the median-EROI assumption, which is 1.0 $GtCO_2$ more than during the 2006–2015 period. Such an increase in emissions represents less than

3% of total carbon emissions in 2020, and does not undermine the target of keeping global warming below 1.5 °C.

Overall, the benefits of rapid decarbonisation far outweigh the extra emissions from the small jump. In pathways of rapid decarbonisation and lower energy use, the increase in emissions due to the upfront energy requirement of low-carbon infrastructure is small. In S1-A, emissions increase by only 0.6 $GtCO_2$ per year from 2020 to 2030 for the median-EROI scenario. Moreover, the phasing-out of fossil fuels leads to a rapid reduction in energy system emissions, starting as early as 2025.

Over the long term, the quantity of emissions depends on the amount of fossil fuels remaining in the energy system and the choice of low-carbon energy technologies in each of the pathways. Energy transitions that rapidly phase-out fossil fuels and prioritise renewables and nuclear energy over bioenergy technologies (BECCS in particular) achieve lower cumulative energy system emissions. The reason is that the emissions associated with renewables converge to zero (as in the LED pathway), while the emissions in pathways with BECCS only level off in the second half of the century (Fig. 3). Pathways that combine an extended use of conventional fossil fuels with BECCS have higher energy system emissions, as they fail to completely decarbonise the energy supply. Moreover, BECCS is a low-EROI technology (see also Supplementary Table 1 and Supplementary Figs. 3, 4). It has a low-energy conversion efficiency, and substantial energy requirements are associated with bioenergy supply[36,37]. As such, BECCS has higher energy system emissions per unit of energy generated when compared to renewables (see Supplementary Fig. 6).

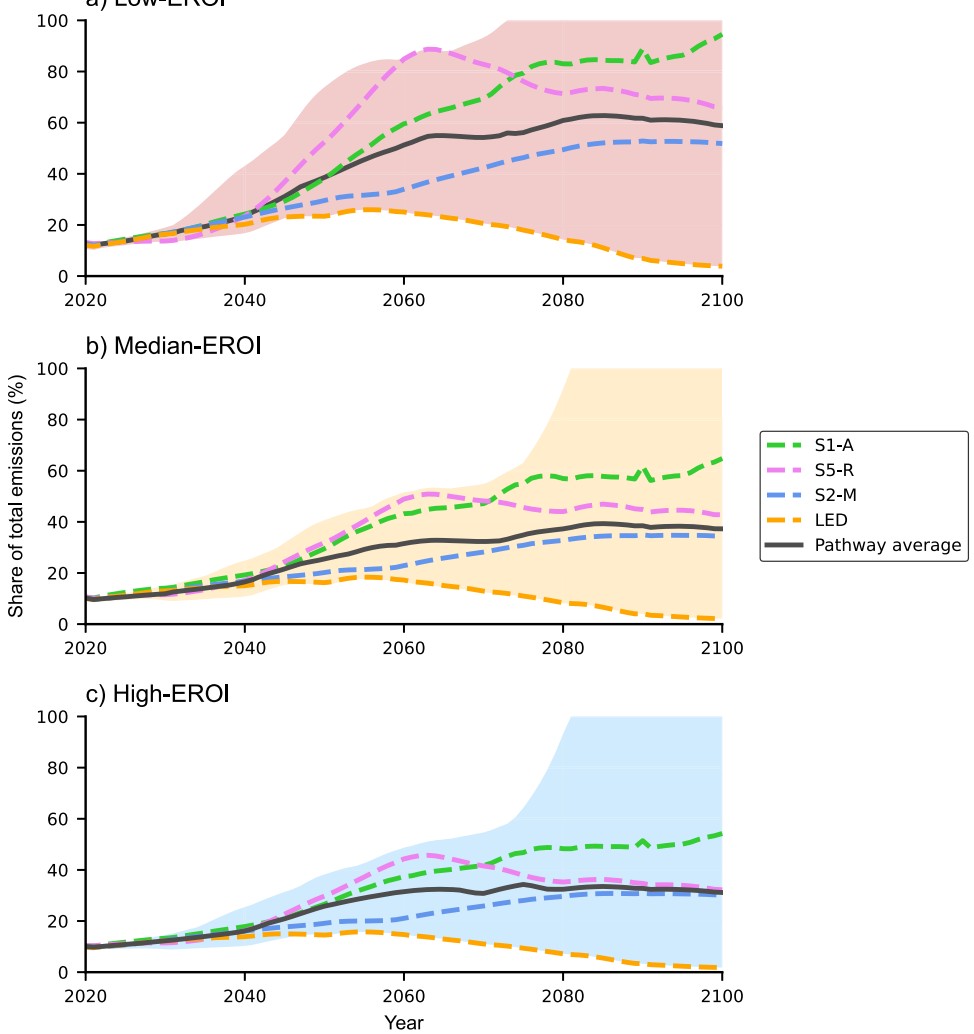

**Fig. 2 | Share of energy system emissions over time, as a percentage of total emissions, for three different EROI scenarios. a** Low, **b** median, **c** high. In each panel, the solid line shows the average result of all fourteen pathways, while the result for each of the four illustrative pathways is plotted as a dashed line. The shaded envelopes show the full range of results for the fourteen pathways. If the values reach 100%, the energy system emissions exceed the total emissions in the respective pathway.

## Avoiding the energy trap

Our findings suggest that a low-carbon energy transition would drive up the share of total energy generation going towards the construction and operation of the energy system, and maintenance of the energy supply, compared to the current energy system. A higher share of energy for the energy system would contribute to a decrease in net energy available to society. Depending on the mitigation pathway, the decrease in per capita net energy could be as low as 10% or as high as 34%.

In pathways of lower energy use and rapid decarbonisation, the increase in the share of energy for the energy system would be largest during the initial push for the transition when the upfront energy requirement to construct low-carbon energy infrastructure would consume an increasing proportion of total final energy (Fig. 4a). In pathways of moderate and slower decarbonisation, the energy share increases in the second half of the century. The average share of energy for the energy system in the fourteen pathways for the median-EROI scenario during the 2020–2030 period is ~14%, with the highest increase occurring in the S1-A pathway, and the lowest in the S2-M pathway (Fig. 4a).

Net energy available to society declines in all of the pathways we analysed, albeit at different rates and over different periods (Fig. 4b). In

the S5-R pathway, the decrease is only temporary. Net energy primarily depends on the growth in final energy and less on changes in the energy for the energy system. Therefore, net energy declines substantially in pathways that increase both the energy for the energy system and reduce final energy. Net energy per capita could drop by 28–34% by 2030, compared to 2015, for pathways of rapid decarbonisation such as S1-A and LED.

Our results are similar to those of King and van den Bergh[12], who estimated a 24–31% reduction in net energy per capita for the IEA low-carbon transition pathway. However, King and van den Bergh's reduction takes place over a longer period (from 2015 to 2050), whereas our results indicate that a low-energy transition could lead to a major reduction in net energy per capita in a single decade (Fig. 4b). In the pathways of slower decarbonisation (S2-M and S5-R), net energy per capita declines later, and by less, with a decrease of 10% by 2040 compared to 2030 in the median-EROI scenario. Net energy in these pathways also only declines temporarily (until 2050), after which it returns to growth.

In contrast to what has been argued in previous studies[9–12,38], we find that a low-carbon energy transition would not necessarily lead to a decline in the EROI of the overall energy system in the long term (Fig. 4c). The EROI of the overall energy system depends on the choice

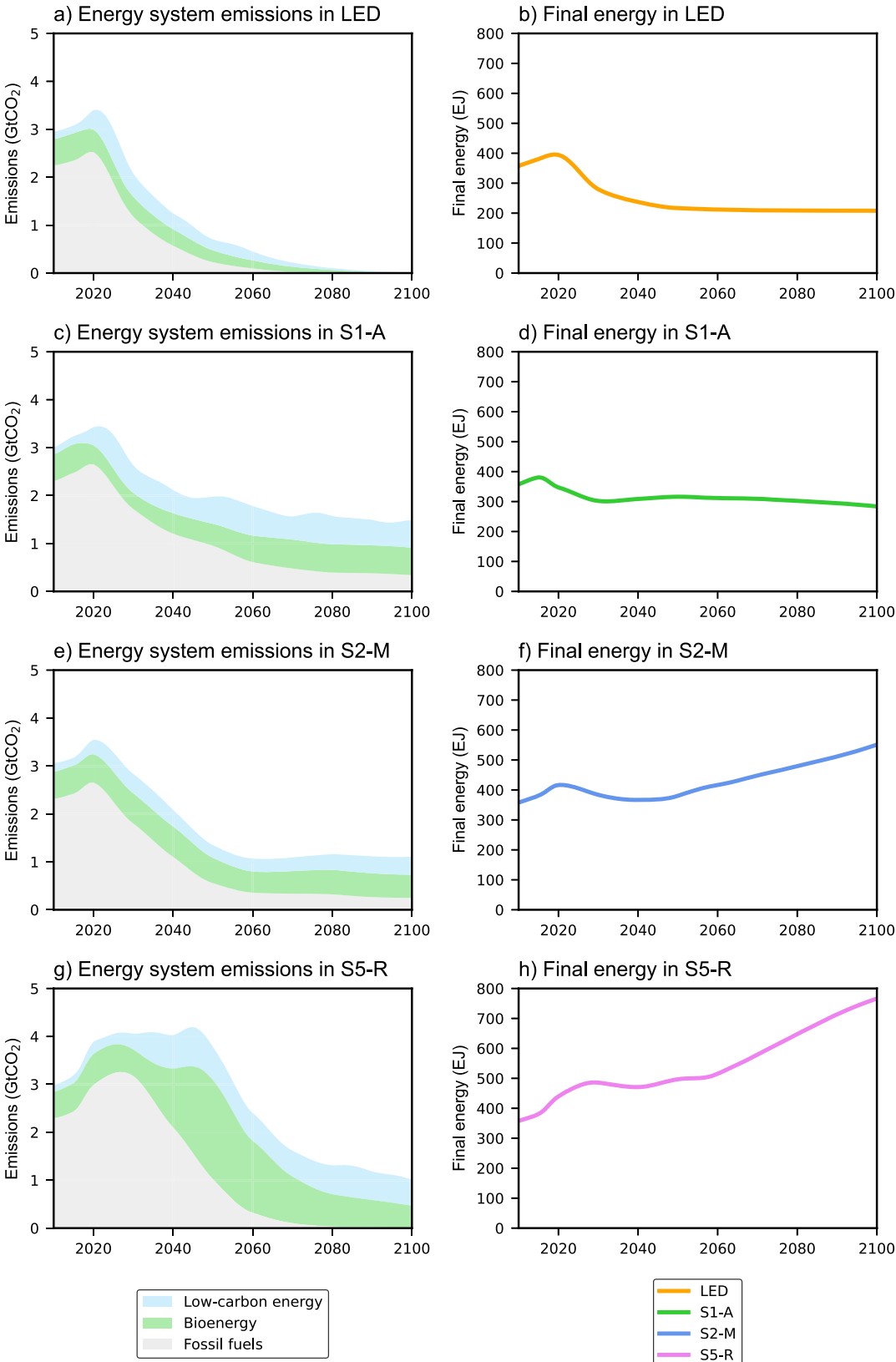

**Fig. 3 | Energy system emissions and energy use in the four illustrative pathways.** The left panels **a**, **c**, **e**, and **g** show annual carbon emissions associated with the energy system under the median-EROI scenario. Energy system emissions are divided between three types of energy conversion technologies: fossil fuels, bioenergy, and low-carbon technologies that include renewables, hydrogen, and nuclear energy. The right panels **b**, **d**, **f**, and **h** show the final energy consumption in the pathways.

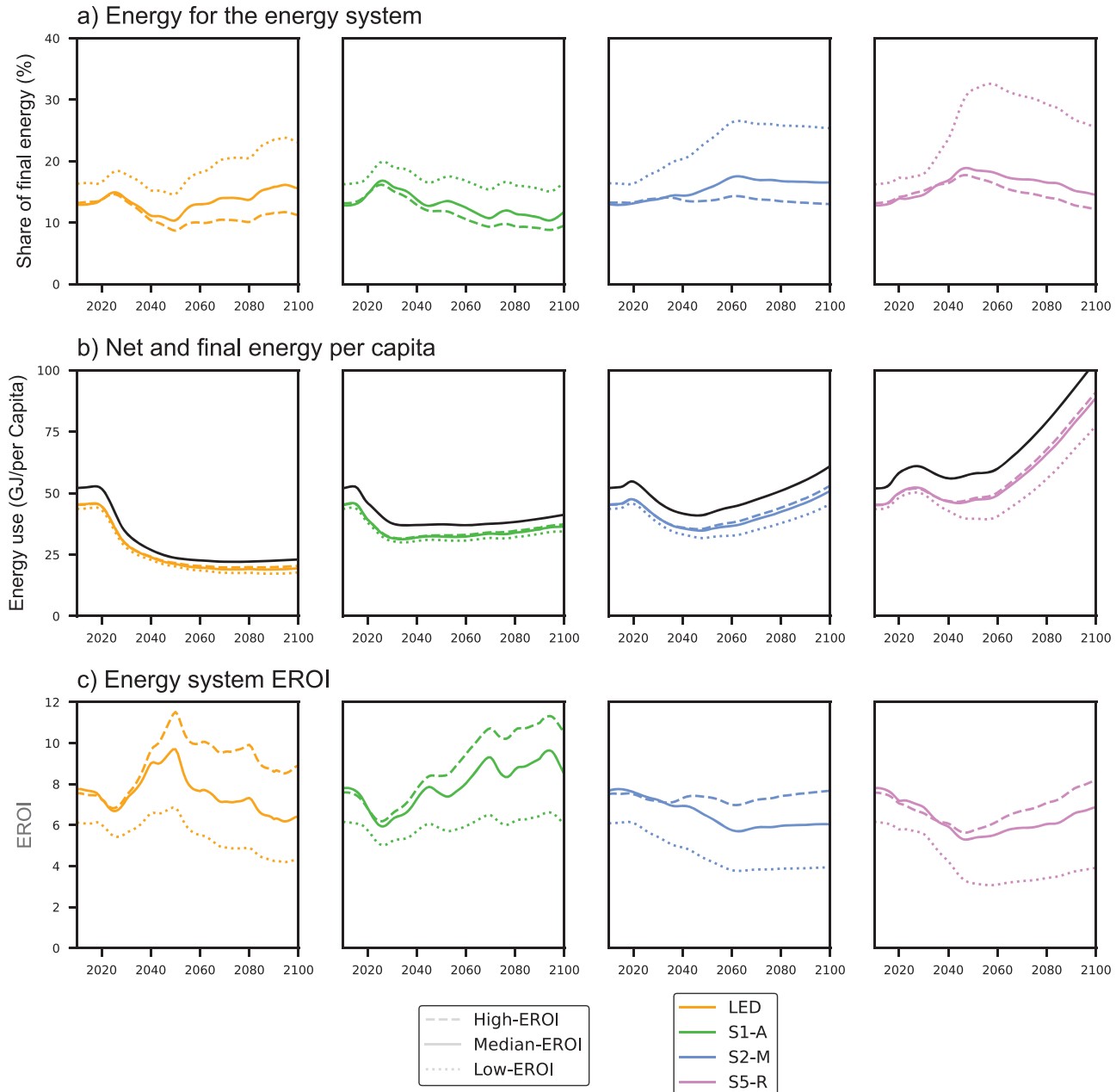

**Fig. 4 | Share of energy for the energy system, net energy per capita, and EROI of the energy system for the four illustrative pathways.** Panel **a** shows the share of final energy that is required for the construction and operation of the energy system. Panel **b** shows final energy per capita (solid black line), and how much of it will be left for society as net energy per capita. Panel **c** shows the evolution of the EROI of the overall energy system. All panels show a range of three estimates: high-EROI (dashed line), median-EROI (solid line), and low-EROI (dotted line).

of energy conversion technologies. EROI declines in pathways that prioritise bioenergy and fossil fuels with carbon capture and storage (e.g. S2-M and S5-R), and increases in pathways that focus on deploying renewable energy technologies (e.g. S1-A). Our results are consistent with the latest findings in the literature, which suggest that the EROI of renewable energy is comparable to (or higher than) the EROI of fossil fuels at present, and likely to increase[16], while the EROI of fossil fuels is likely to decrease[14]. However, pathways, where EROI is likely to decline, can still provide more net energy to society by increasing energy production, even though they require more energy to support the energy system (e.g. S5-R).

All pathways suggest an inevitable decline in per capita net energy at some point during the transition. However, this finding does not mean that energy scarcity is an unavoidable feature of any low-carbon energy transition. The projected net energy decline is not due to constrained possibilities of energy growth in the models, but because the models assume more efficient energy use, which makes such pathways cost-effective[31].

The prospect of more efficient energy use in society means that fundamental energy services such as heating, lighting, and transportation could still be provided even if less net energy were available. Access to fundamental energy services could be maintained in high-income countries, and increased in lower-income countries, at much lower net energy levels[39–41]. A good life could be achieved at lower per capita energy use by improving the efficiency of energy using technologies (e.g. by replacing gasoline-powered cars with electric cars), by shifting from consumption choices with higher energy intensities to choices with lower energy intensities (e.g. from cars to bicycles), and by avoiding the most inefficient alternatives altogether (e.g. flying)[42].

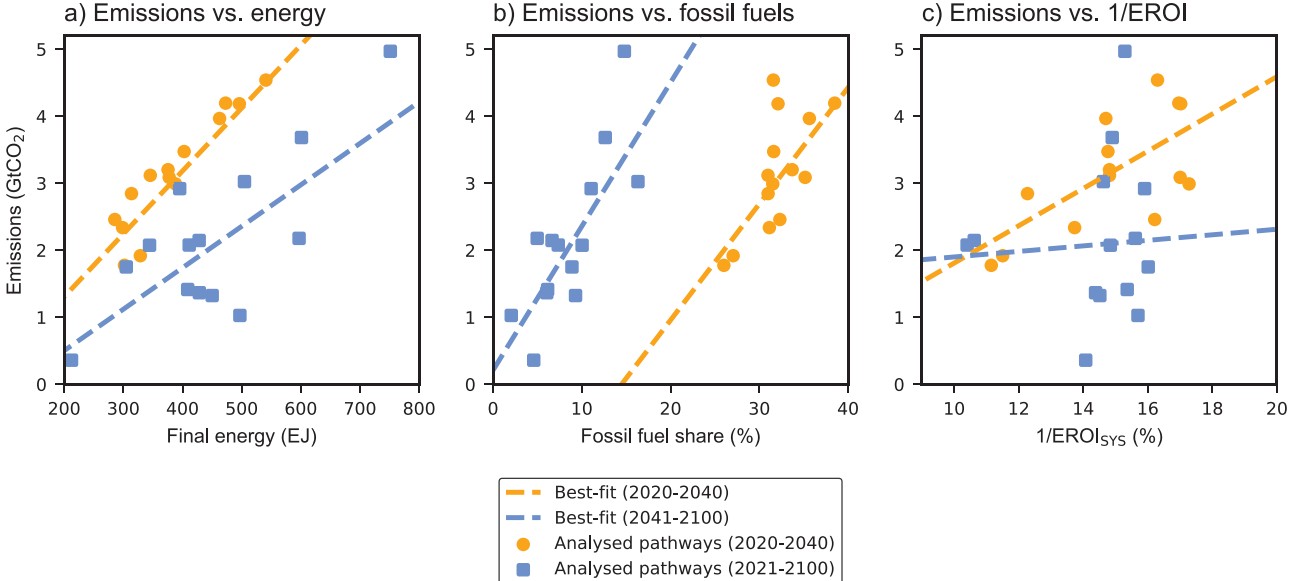

**Fig. 5 | Analysis of factors affecting energy system emissions.** The figure shows the relationship between average energy system emissions and different factors for each of the fourteen pathways compatible with 1.5 °C, for the median-EROI assumption. Panel **a** shows average energy system emissions in relation to average annual energy use during the initial push for transition (2020–2040), and the period following this push (2041–2100). Panel **b** shows average energy system emissions in relation to the average share of energy generated from conventional fossil fuels, over the two periods. Panel **c** shows average energy system emissions as a function of the average share of energy for the energy system for the two periods.

## Factors driving energy system emissions

Our results suggest that energy system emissions are substantial, and increase as a share of total emissions over time. We use a panel data analysis (Supplementary Table 9) to analyse the underlying factors behind energy system emissions during the transition while controlling for heterogeneity across the fourteen analysed pathways over time. Our analysis shows that energy system emissions depend on the growth in final energy use and the choice of energy technologies during the transition. A decrease in the overall EROI of the energy system contributes to an increase in energy system emissions during the initial push for the transition (from 2020 to 2040) but does not have a clear effect on emissions thereafter. The pathways that provide more energy to society have higher energy system emissions.

The different relationships in the two periods can be seen in Fig. 5. The relationship between energy use and energy system emissions is particularly strong during the initial push for the transition (as shown by the orange markers in Fig. 5a). From 2020 to 2040, a 100 EJ increase in annual energy use is associated with a 0.8 GtCO$_2$ increase in annual energy system emissions. The relationship weakens after 2040 as the energy system is gradually decarbonised (see blue markers in Fig. 5a). From 2041 to 2100, the share of energy from fossil fuels is the most important factor contributing to energy system emissions (Fig. 5b and Supplementary Table 9).

In theory, energy system emissions could be decoupled from the scale of the energy system completely by fully substituting fossil fuels with energy from renewables and nuclear energy[5]. However, such a transition would require even more dramatic upscaling of these technologies than currently assumed in the IPCC literature[43]. More-over, this upscaling could be constrained by other factors, such as the supply of materials required for energy infrastructure[44,45]. Such issues may be best addressed by improved models that explicitly calculate the energy and material requirements of the transition, beyond what is covered by existing IAMs.

Finally, we find a weak relationship between the share of energy for the energy system (defined as 1/EROI$_{SYS}$) and energy system emissions from 2041 to 2100. During the latter years of the transition, the overall EROI of the energy system becomes a secondary factor for

emissions (as shown by the wide scatter in the blue markers in Fig. 5c). This is not to say that EROI is not a relevant factor, as a lower EROI means that the energy system requires a larger share of total energy. However, a lower EROI may be counterbalanced by a lower share of fossil fuels in the energy system (e.g. due to faster decarbonisation), or by lower energy use.

## Discussion

In this article, we have calculated the energy for the energy system, and the corresponding energy system emissions, for fourteen 1.5 °C–compatible mitigation pathways used extensively in the IPCC literature. Although energy for the energy system and energy system emissions are implicitly accounted for in these pathways, they are not quantified and reported as separate quantities. By providing a separate picture of energy system emissions, we complement existing IAMs, yielding three core insights.

First, we find that energy system emissions are substantial. On average, we estimate that energy system emissions for a low-carbon transition would amount to 195 GtCO$_2$, which corresponds to ~5 years of global CO$_2$ emissions at their 2021 level. Based on the modelled linear relationship between total cumulative emissions and global warming[46], this figure implies that a low-carbon energy transition would lead to approximately 0.1 °C of additional global warming. Therefore, although the cumulative energy system emissions are substantial, their overall climate impact is small compared to the amount of carbon saved over the long term by rapid decarbonisation.

Second, we do not find a large jump in energy system emissions in the short run from intensifying efforts to decarbonise the energy system. On the contrary, we find energy system emissions to be higher in pathways that decarbonise slowly, that use more fossil fuels to produce energy in the short term, and that rely on negative emission technologies to compensate for higher cumulative emissions (Fig. 3 and Supplementary Fig. 6). Contrary to previous concerns about emissions associated with a transition to renewable energy increasing emissions in the short run, we identify a longer-term problem in pathways of slower decarbonisation and large-scale carbon removal, as energy system emissions in these pathways continue well into the

future (Fig. 3). Although modest on their own, these emissions are comparable in magnitude to the residual emissions from aviation, steel and cement production, and load-following electricity. Our results complement studies that find that carbon removal by BECCS is a much less efficient mitigation approach than assumed in existing pathways, due to upstream emissions from biomass supply chains[36,47] and land-use change[48,49].

Third, we find a comparable reduction in net energy, and in the share of energy available to society, during the low-carbon transition to that found in previous studies[12]. However, reductions in our study tend to come earlier (within the first decade of efforts), especially for mitigation pathways of fast decarbonisation. Pathways with faster decarbonisation and lower energy demand have lower energy system emissions, but this comes at the cost of lower net energy for society. Lower net energy does not need to lead to energy scarcity. A consensus is emerging regarding the enormous potential to use energy more efficiently[50,51], and the possibilities of providing a decent life with much less energy than is currently consumed in wealthy nations[39,41,52].

In general, our study demonstrates the importance of calculating the energy requirements and emissions associated with the transition, to get a more complete picture of energy system dynamics and to quantify the remaining emissions available to society. Further research could explore the energy required and emissions associated with the replacement of machines and infrastructure at the consumption end of the energy system (e.g. electric vehicles, their charging stations, and energy storage solutions). Calculating such emissions would be worthwhile, as the transformation of the consumption end of the energy system could potentially take up a large part of the remaining carbon budget for 1.5 °C.

The research on energy transitions should go beyond the scenarios produced by IAMs, and also include scenarios from alternative "normative" energy modelling approaches[5,43,53]. IAMs focus on optimal-cost pathways of decarbonisation, and therefore do not cover the whole range of possible energy transitions[54]. IAMs have been found to be biased towards technologies that are direct substitutes for conventional fossil fuels, such as BECCS and fossil fuels with carbon capture and storage[55], which is why they tend to underestimate the realistic deployment potential of intermittent renewables[56].

In our analysis, we find that a preference for direct substitutes for conventional technologies leads to higher energy system emissions and lower net energy. A discussion of whether a low-energy transition based on renewables would be a preferable mitigation strategy is beyond the scope of this article. However, our analysis suggests that explicit modelling of energy system emissions and dynamic EROIs from different energy technologies could add support to the case for renewables over technologies relying on carbon capture and storage.

Questions remain regarding the extent to which the production-based approach of IAMs accounts for the upstream emissions associated with different energy generation technologies[21], and also the extent to which IAMs capture effects from changes in the EROI of the energy mix[57]. Further research should explore the possibility of integrating EROI analysis into IAMs to produce internally more consistent energy and emissions pathways, which would likely change the models' choice of energy generation technologies. Such integration could involve EROI scenarios tailored to the narratives of the mitigation pathways, and link EROI calculations to specific narrative assumptions (e.g. about technological change, international cooperation, land-use, and innovation).

Overall, our study demonstrates the importance of accounting for net energy and energy system emissions. Future mitigation pathways would be improved by explicitly modelling the energy requirements and emissions associated with a low-carbon energy transition. Doing so would allow us to better understand the trade-off between the energy and carbon required to transition to a low-carbon energy

system, and what remains for other socioeconomic activities outside of the transition.

## Methods
### Energy transition pathways
Fourteen 1.5 °C–consistent (RCP1.9) mitigation pathways were selected for this study. For illustrative purposes, we focus on four pathways (LED, S1-A, S2-M, and S5-R) from the IPCC's Special Report on Global Warming of 1.5 °C, which the IPCC selected as illustrative archetypes of alternative low-carbon transitions[1]. These transitions are model interpretations of four distinct narratives that describe possible socio-economic and technological developments in a world that limits climate change to 1.5 °C. The narratives are: Low Energy Demand (LED)[31], Sustainability (S1)[58], Middle of the Road (S2)[59], and Fossil-fuelled Development (S5)[60].

To capture a wider range of assumptions and modelling frameworks beyond the four illustrative pathways, we complement the analysis with ten additional pathways, which were produced in the same study as S1-A, S2-M, and S5-R by Rogelj et al.[30]. Five of these are modelling representations of S1, three of S2, and one each of S4 and S5. S4 is also known as the "world of deepening inequality" narrative[61].

### Energy requirements and EROI
We estimate the energy for the energy system during transition by applying the analytical framework of Energy Return on Investment (EROI) at the final energy stage. EROI describes a ratio between the amount of net energy delivered to society ($E_{NET}$) and the total amount of energy that is required to extract, convert, and deliver this energy ($E_{REQ}$), which we also refer to as the energy for the energy system[23,62]. EROI is a measure of energy system efficiency, as it compares the amount of energy that enters the productive economy with the energy that is associated with total (gross) energy production[14,63]. The lower the EROI, the greater the energy requirements, and the lower the net energy that is available for productive socioeconomic activities (see Eq. 1).

$$\text{EROI} = \frac{E_{NET}}{E_{REQ}} = \frac{E_{GROSS} - E_{REQ}}{E_{REQ}} = \frac{E_{GROSS}}{E_{REQ}} - 1 \qquad (1)$$

We define the system boundaries of our EROI analysis at the final energy boundary, also known as the point-of-use boundary, which describes the point where energy carriers enter the productive economy[64]. $\text{EROI}_{FIN}$ includes all of the direct inputs along the energy supply chain required to extract ($E_{EXT}$) and refine energy resources ($E_{REF}$), the energy used to transport the energy from the primary energy stage to the point of use for society ($E_{TRA}$), as well as the energy requirements associated with construction ($E_{CON}$), decommissioning ($E_{DEC}$), and operation and maintenance of energy infrastructure ($E_{O\&M}$), such as power plants and refineries, as shown in Eq. 2[24]:

$$E_{REQ} = E_{EXT} + E_{REF} + E_{TRA} + E_{CON} + E_{O\&M} + E_{DEC} \qquad (2)$$

Energy for construction refers to the energy that is used to manufacture and build energy infrastructure like power plants and refineries. Energy for decommissioning accounts for the energy required to dismantle, remove, and dispose of obsolete energy infrastructure. Energy for operation and maintenance includes energy used to extract primary energy resources, and the energy required to convert primary energy into useful energy carriers and deliver them to the end-user. Energy for operation and maintenance also includes all of the energy inputs for the energy industry's own use, from the primary to the final energy stage.

By convention, energy conversion losses from primary to final energy and energy losses in distribution, transmission, and storage ($E_{LOSS}$) are not counted among the energy requirements of energy

conversion technologies[10]. These losses are already accounted for in the energy balances of the original data from the pathways and result in lower final energy relative to total energy generation. Moreover, the energy requirements do not include the raw energy embodied in energy resources (e.g. the heating value of gas) that are to be converted into useful carriers. The energy requirements only account for energy inputs that are needed to procure and process the resources into useful energy carriers, and to deliver these carriers to the end-user. See Supplementary Fig. 7 for a complete illustration of our energy system boundaries and the representation of energy flows from primary energy sources to net energy.

We estimate the energy requirements for twenty-seven energy conversion technologies that are represented in the mitigation pathways (see Supplementary Figs. 1 and 2 for a detailed overview of all energy conversion technologies in our model). These technologies describe different pathways of energy conversion from fossil fuels and biomass alongside energy generated from non-biomass renewables, nuclear energy, and hydrogen. In our calculations of energy for the energy system, we distinguish between four types of energy carriers that are represented in the mitigation pathways: electricity, refined liquid fuels, gases, and solids (coal and combustible biomass).

## Energy requirements of fossil fuel and biomass technologies

For energy conversion from fossil fuels and biomass, the main energy requirements are associated with the extraction, processing, and delivery of energy resources, whereas the construction, decommissioning, and operation and maintenance of the energy infrastructure represent only a small share of total energy requirements[25]. By contrast, for non-biomass renewables, almost all energy requirements are from upfront energy demand for the construction of energy infrastructure.

We estimate the energy requirements associated with construction as upfront energy invested during the first year of the energy facility's lifetime. Similarly, the energy required for decommissioning is accounted for at the end of the energy infrastructure lifetime. The remaining energy inputs that are associated with energy system operations are counted every year during the lifetime of the energy infrastructure.

To calculate the energy requirements to build, decommission, and operate and maintain the energy infrastructure, we follow the previous work of Sgouridis et al.[65]. We calculate the energy required for the construction and the energy embodied in the energy generation machinery by estimating the energy intensity of capital ($\varepsilon$) and multiplying it first by the capital costs of infrastructure per unit of installed power ($C_p$), and second by the newly installed power capacity in the respective year ($P_{NEW}$), as shown in Eq. 3. For infrastructure capital costs, we use values from the REMIND IAM documentation[66], which are provided in \$US2015. We estimate the energy intensity of capital at 4.52 TJ/million \$US2015, after adjusting for inflation the estimate of 5.49 TJ per million \$US2007 from the abovementioned study, using the producer price index from the PCU3336 industry group data[67]. Values of the parameters for different energy conversion technologies are listed in Supplementary Table 10.

$$E_{CON}(t) = \varepsilon \times C_p \times P_{NEW}(t) \tag{3}$$

In our calculations of energy requirements associated with new energy infrastructure, we include the power capacity built to increase energy production as well as the capacity that replaces the infrastructure that is decommissioned at the end of its lifetime ($\tau$), as shown in Eq. 4:

$$P_{NEW} = \begin{cases} \max(0, P(t) - P(t-1)); t < \tau \\ \max(0, P(t) - P(t-1) + P_{NEW}(t-\tau)); t \geq \tau \end{cases} \tag{4}$$

We calculate the energy for the operation and maintenance of energy infrastructure as a product of the energy intensity of capital and the operation and maintenance costs per unit of generated energy ($C_{O\&M}$) multiplied by the total energy generated per year ($E_{GEN}$), as shown in Eq. 5:

$$E_{O\&M}(t) = \varepsilon \times C_{O\&M} \times E_{GEN}(t) \tag{5}$$

In estimating the energy required for the decommissioning of energy infrastructure at the end of its life we apply the assumption of Hertwich et al.[35], who estimates that decommissioning represents roughly 10% of the energy required for construction (see Eq. 6).

$$E_{DEC}(t) = 0.1 \times E_{CON}(t - \tau) \tag{6}$$

In the following steps, we describe the calculation of the energy requirements of processes for obtaining raw fuels before they are refined into useful energy carriers that can be delivered to end-users.

To estimate the energy used in the extraction, mining, or harvesting of raw fuels, we collect a series of present-day EROI estimates at the standard energy system boundary[28] (e.g. farm-gate or mine-mouth; denoted $EROI_{ST}$) from the peer-reviewed literature, as listed in Supplementary Table 11. From the $EROI_{ST}$ values of these selected studies, we calculate the lower, median, and upper inter-quartile range of the $EROI_{ST}$ for each energy resource and use these values to determine a range of estimated energy requirements associated with the appropriation of raw energy fuels. We assume the $EROI_{ST}$ of fossil fuels will continue to decline over time. We model the decline by following the approach of Dale et al.[29] and Sgouridis et al.[5], who use the equation of exponential decline from present-day values $EROI_{ST}(0)$ shown in Eq. 7. This approach models the convergence of $EROI_{ST}$ towards the minimum EROI ($EROI_{ST,low}$), which we assume corresponds to the lower inter-quartile range of present-day EROI estimates. The rate of decline ($\beta_C$) for each respective resource is calibrated from the historical trend for the $EROI_{ST}$ of fossil fuels, as published by Brockway et al.[14].

$$EROI_{ST}(t) = EROI_{ST,low} + (EROI_{ST}(0) - EROI_{ST,low}) \times exp^{-\beta_C \times t} \tag{7}$$

$EROI_{ST}$ compares the raw energy content of energy resources such as wood, coal, gas, and crude oil ($E_{RAW}$) with the energy required to obtain these fuels ($E_{EXT}$; see Eq. 8), before they are converted into useful energy carriers. The efficiency of energy conversion ($\eta_C$) depends on the respective energy conversion technology and may change over time. In this study, we apply the energy conversion coefficients from the representation of energy technologies in the REMIND model[66,68]. The model assumes energy conversion efficiency in new energy infrastructure improves over time. We combine Eq. 8 and 9 to obtain an expression that links the energy requirements of extraction (or harvest or mining) to the efficiency of energy conversion and $EROI_{ST}$ (Eq. 10).

$$EROI_{ST}(t) = \frac{E_{RAW}(t)}{E_{EXT}(t)} \tag{8}$$

$$E_{GEN}(t) = \eta_C(t) \times E_{RAW}(t) \tag{9}$$

$$E_{EXT}(t) = \frac{E_{GEN}(t)}{\eta_C(t) \times EROI_{ST}} \tag{10}$$

In estimating the energy required for the refining or processing of fuels ($E_{REF}$) we refer to the calculations from previous studies. For the refining of crude oil, we use the estimates of energy intensity of refining in MJ per kg ($\mu_{REF}$) from Raugei and Leccisi[63] and the "Ecoinvent Life-cycle Inventories of Oil Refinery Processing"[69]. For the

processing of raw fuels from biomass, we use estimates of energy intensity from an extensive literature review by Fajardy et al.[36,37]. We define the energy used in refining as a product of the mass of the respective fuel and the energy intensity of refining, as shown in Eq. 11. We calculate the mass of the fuel by dividing the raw energy content of energy resources ($E_{RAW}$) by the higher heating value (HHV), described by Eq. 12. We do not assume specific energy requirements for the processing of natural gas and coal, consistent with previous EROI and life-cycle studies[63,70].

$$E_{REF}(t) = M_{FUEL}(t) \times \mu_{REF} \tag{11}$$

$$M_{FUEL}(t) = \frac{E_{RAW}(t)}{HHV} \tag{12}$$

To calculate the energy requirements for transportation, we assess global trade routes of coal, gas, and crude oil in the year 2019, by using the flows of these fuels from the international trade balance sheets of the BP Statistical Review of World Energy 2020. For biomass, we use data on the global flows of wood pellets from Junginger et al.[71]. We partition the trade routes (indexed with l) into different stages by transportation type, estimating the average trade distance in each route. For example, the oil route from Baghdad (Iraq) to Houston (USA) consists of an onshore pipeline of 970 km from Baghdad to Ceyhan (Turkey), a sea freight route of 12,500 km from Ceyhan to Houston, and an onshore pipeline of 100 km on the US mainland. We assume that the energy intensities of fuel transportation types remain constant over time.

We calculate the energy used in each stage of transportation (indexed with j) by multiplying the amount of fuel transported by the energy intensity of the transportation type and the distance over which the fuel is transported, as shown in Eq. 13. The parameters for the transportation types are obtained from the life-cycle inventory database EcoInvent v3.2[72], and can be found among the parameters listed in Supplementary Table 10.

$$M_{FUEL,l} \times \gamma_{TRA,j} \times distance_{l,j} \tag{13}$$

To estimate the average global energy intensity ($\epsilon_{TRA}$) associated with the transportation of each fuel (in MJ/kg), we sum the energy use across the global trade routes and divide the sum by the global volume of trade flows (in tonne kilometres), defined as the global sum of transported fuel multiplied by the distance, as described in Eq. 14:

$$\epsilon_{TRA} = \frac{\sum_l \left( M_{FUEL,l} \times \sum_j \gamma_{TRA,j} \times distance_{l,j} \right)}{\sum_l \left( M_{FUEL,l} \times \sum_j distance_{l,j} \right)} \tag{14}$$

Finally, we obtain the energy required for the transportation of raw fuel by multiplying the mass of the fuel transported by the average global energy intensity of fuel transportation for each respective fuel, as shown in Eq. 15:

$$E_{TRA}(t) = M_{FUEL}(t) \times \epsilon_{TRA} \tag{15}$$

For a complete overview of our assumptions regarding the trade routes of coal, natural gas, crude oil, and biomass, and our calculations of the energy intensities of fuel transport, see Supplementary Tables 12–15.

## Energy requirements of non-biomass renewables and nuclear energy

The largest energy requirements of non-biomass renewables (i.e. solar photovoltaics, wind, geothermal, and hydropower) are related to the manufacturing and construction of energy infrastructure[25,35]. For renewables, the energy required for operation is much lower than technologies that produce energy from raw fossil fuels, as renewable sources do not require energy to be extracted, transported, and processed. For nuclear energy, the energy to maintain the energy supply chain also includes energy requirements for the extraction, enrichment, and transportation of uranium. Here, the energy requirements for operating energy infrastructure and maintaining the energy supply are substantially higher compared to the construction of energy infrastructure.

To obtain estimates of the energy requirements of renewables and nuclear energy over the lifetime of each technology, we collected a series of present-day EROI estimates for each technology at the final energy boundary, from a number of peer-reviewed studies (see the studies listed in Supplementary Table 1). From these studies, we calculated the lower, median, and upper quartiles of the range of EROI values for each energy source. These quartiles are classified as low, median, and high-EROI estimates.

We divided the energy requirements between the energy required for construction ($E_{CON}$) and decommissioning ($E_{DEC}$) of the energy infrastructure, and the annual energy requirements to operate the energy infrastructure and maintain the energy supply ($E_{O\&M}$), following the approach of King and van den Bergh[12]. Energy requirements for operation are proportional to the total installed power ($P$) times the capacity factor (CF) divided by the EROI of the technology[12], as shown in Eq. 16. CF is a dimensionless ratio that compares the actual annual generation of energy to the maximum potential energy output. The parameter $\alpha_{tech}$ describes the ratio between the energy requirements of operation and the energy invested in construction over the lifetime of the technology.

$$E_{O\&M}(t) = \frac{\alpha_{tech} \times P(t) \times CF(t)}{EROI(t)} \tag{16}$$

As described in Eq. 17, the energy requirements of construction are proportional to newly installed power ($P_{NEW}$), times the capacity factor of the respective energy conversion technology (CF), multiplied by the lifetime of the technology ($\tau$), and divided by the technology's EROI.

$$E_{CON}(t) = \frac{(1 - \alpha_{tech}) \times P_{NEW} \times CF \times \tau}{EROI(t)} \tag{17}$$

Energy associated with decommissioning is assumed to represent 10% of the energy used for construction, following Hertwich et al.[35] Energy associated with decommissioning is accounted for in the last year of the energy infrastructure's lifetime, as shown in Eq. 18:

$$E_{DEC}(t) = 0.1 \times E_{CON}(t - \tau) \tag{18}$$

We assume the historical trend of increasing EROIs of photovoltaic and wind power technologies will continue in the future. We model the EROI dynamics of these technologies by applying "energetic experience curves" (see Steffen et al.[16] and Loueven et al.[17]), thus estimating the reduction in the energy requirements for construction, and operation and maintenance due to technological innovation. For a detailed explanation of how we calculated the future dynamics of EROI for photovoltaics, wind power, and hydrogen from electrolysis, see the "Note on EROI dynamics of wind and solar power" and the "Note on energy requirements for hydrogen from electrolysis" in the Supplementary Information. In estimating the energy requirements of

hydropower, geothermal, and nuclear energy, we refer to the present-day range of EROI estimates, due to a lack of studies on EROI dynamics for these technologies.

## EROI of the overall energy system

We calculate the EROI of the overall energy system at the final energy stage, by applying Eq. 1, wherein we compare the total amount of gross final energy production to the sum of the energy requirements for all energy conversion technologies (here represented by the index $i$), as shown in Eq. 19. We use the same approach to calculate the EROI of individual energy conversion technologies and the EROI of different carriers, such as the EROI of electricity from renewables.

$$\mathrm{EROI}_{\mathrm{SYS}} = \frac{E_{\mathrm{GROSS}}}{\sum_i E_{\mathrm{EXT},i} + E_{\mathrm{REF},i} + E_{\mathrm{TRA},i} + E_{\mathrm{CON},i} + E_{\mathrm{O\&M},i} + E_{\mathrm{DEC},i}} - 1 \quad (19)$$

We test our model by comparing our estimates of the EROI of the overall energy system with the results from the EROI literature. The note on "EROI estimates of different energy carriers" in the Supplementary Information demonstrates that our calculations of the EROI at the final energy stage are consistent with estimates from previous studies.

EROI values differ greatly depending on the energy system boundaries that the analyst uses[22,64]. For example, some studies measure energy delivered at the point of energy extraction, while others calculate energy delivered to the end-user, which is an expanded analytical boundary of the system. Expanding the boundary results in lower EROI values, as it includes the additional energy required to convert the raw resource into useful energy and move or store it. We selected studies to match a consistent system boundary, which includes the energy investments for energy resource extraction, resource transportation, resource processing, the construction of energy conversion facilities, and the energy required for the operation of the facilities.

We assume a global average EROI for each energy conversion technology. We do not take into account regional differences in production and transformation processes[28]. However, the EROI of the entire energy system does change with improvements in energy conversion efficiencies, changes in the $\mathrm{EROI}_{\mathrm{ST}}$ of fossil fuels due to a declining abundance of these energy resources, and as the mix of energy technologies change over time.

In our EROI scenarios, high-EROI values assumed for each energy technology are based on studies with favourable assumptions regarding resource abundance and deployment of the most efficient low-carbon energy-generating technologies. Low-EROI values, in turn, assume lower resource abundance and limited technological improvement of low-carbon energy technologies. Median-EROI values represent a balanced, middle-of-the road EROI trajectory. For a detailed overview of EROI assumptions for different energy technologies, see Supplementary Tables 3–7.

## Net energy

To calculate net energy per capita, we divide the difference between gross final energy and the total energy requirements of the energy system (as shown in Eq. 1), by the global population projections in the mitigation pathways.

## Energy system emissions

Estimating the carbon emissions associated with the build-up of the energy system and the operation and maintenance of the energy supply during transition is crucial for assessing different mitigation pathways. If a substantial amount of the remaining carbon budget goes to decarbonising the energy system, this may significantly affect the projections of energy use and emissions in the end-use energy sectors. Future energy system emissions depend on changes in the energy

requirements of the energy system and the carbon intensity of the energy for the energy system. Energy system emissions decrease with the decarbonisation of the energy supply and a reduction in energy requirements.

To calculate the emissions associated with the construction of energy technologies, and the operation and maintenance of the energy supply over time, we first separate the energy requirements associated with the construction, decommissioning, and operation and maintenance of the energy supply for each energy technology into the four energy carriers: electricity, gases, liquid fuels, and solids. This step is crucial to adequately quantify energy system emissions, as the carbon intensities of different energy carriers can differ substantially[21], especially given that electricity can be decarbonised much faster than other carriers. We count hydrogen among the liquid fuels, assuming that most hydrogen will be destined to replace liquid fossil fuels. In decomposing the energy requirements into different energy carriers, we follow the approach of Arvesen et al.[25], who distinguish between the four abovementioned energy carrier types, for each of the three life-cycle assessment phases of construction, decommissioning, and operation (see Eq. 20). The life-cycle phase of operation and maintenance includes both the energy requirements to operate the energy infrastructure as well as the energy required to maintain the energy supply. We use the life-cycle assessment database from Arvesen et al. to decompose the energy requirements into four energy carriers by multiplying the total energy requirements by the vector of the respective energy carriers shares, composed of electricity (e), gases (g), liquid fuels (l), and solids (s):

$$\mathbf{E}_{\mathbf{REQ},i}(t) = E_{\mathrm{REQ},i}(t) \times \langle e,g,l,s \rangle_i \quad (20)$$

Energy requirements, decomposed into four energy carriers, are multiplied by the carbon intensity vector containing the carbon intensities of energy carriers (**CI**), to obtain the energy system emissions from each respective energy generation technology, and the life-cycle phase, as shown in Eq. 21.

$$\mathrm{CO}_{2,i}(t) = \mathbf{E}_{\mathbf{REQ},i}(t) \bullet \mathbf{CI}(t) \quad (21)$$

We calculate the carbon intensity of each carrier (c) by dividing the total carbon emissions from energy generation for each carrier by the total amount of energy generated by each carrier, as shown in Eq. 21. Changes in the carbon intensities of energy carriers in the mitigation pathways over time are depicted in Supplementary Figs. 8 and 9.

$$\mathrm{CI}_c(t) = \frac{\sum_i \mathrm{CO}_{2,i,c}(t)}{\sum_i E_{\mathrm{GEN},i,c}(t)} \quad (22)$$

Emissions from electricity generation are obtained directly from the original scenario data, whereas emissions from gases, liquids, and solids are calculated using the carbon intensities of energy conversion technologies ($\varphi_i$), which are endogenous to the REMIND model (see Eq. 23). For an overview of the carbon intensity parameters, see Supplementary Table 10.

$$\mathrm{CO}_{2,i,c}(t) = \varphi_i \times E_{\mathrm{GEN},i,c}(t) \quad (23)$$

Cumulative energy system emissions are calculated as the sum of annual emissions from all of the energy generation technologies, over the period from 2020 to 2100. The share of energy emissions that is shown in Fig. 2 is calculated by dividing energy system emissions by the total carbon emissions from energy and industrial processes.

We report negative emissions, realised by BECCS technologies, separately from (positive) anthropogenic emissions. Negative

emissions realised in energy generation are therefore not counted in the calculation of the carbon intensities of the four energy carriers. We refer to the negative emissions data, as they are reported in the original scenario data. Mitigation pathways report the total amount of sequestered carbon by BECCS technologies (gross negative emissions), but do not separately report the positive emissions from BECCS. Positive emissions from BECCS include the emissions from land-use change, the emissions from fertilisers, the emissions associated with the construction and operation of the BECCS energy facilities, and the carbon emitted along the biomass supply chain[73].

Mitigation pathways use different reporting methodologies for the carbon removal by BECCS that, in some cases, combine gross carbon removal with removals in the land-use sector[74]. This makes it difficult to include the total positive emissions associated with BECCS in our energy system calculations. Here, we limit our analysis of energy system emissions to the emissions associated with the biomass supply and emissions from BECCS facilities, though emissions from land-use-change and fertiliser use may be an even bigger source of energy system emissions[47,48]. As a result, our estimates may considerably underestimate the energy system emissions in low-carbon energy transitions that assume a large-scale use of bioenergy, such as the S2 and S5 mitigations pathways.

### Multiple regression panel data analysis

To quantify the factors driving energy system emissions, we selected three independent variables: final energy use, the share of energy from conventional fossil fuels, and the share of energy for the energy system. Panel OLS multiple regression analysis was used to estimate the contribution of each of these factors. The estimated model is as follows:

$$CO_{2,k}(t) = \gamma(t) + \beta_1 x_{1,k}(t) + \beta_2 x_{2,k}(t) + \beta_3 x_{3,k}(t) + \varepsilon_k(t) \quad (24)$$

where $CO_{2,k}(t)$ is the energy system emissions for pathway $k$ in year $t$, $\beta$ gives the coefficients for the three independent variables $x$, and $\gamma$ is the time-specific term that controls for unobserved heterogeneity over time, and $\varepsilon_k$ is the error term. Time fixed-effects were included in the model, given that we are interested in how the relationship between independent variables and energy system emissions varies between different pathways. Robust standard errors controlling for heteroskedasticity and autocorrelation were estimated after testing for their presence in the balanced panel dataset.

## Data availability

Energy, capacity, emissions, and population data for energy transition pathways were obtained from the IAMC 1.5 °C Scenario Explorer repository (release 2.0)[75], available at: https://data.ene.iiasa.ac.at/iamc-1.5c-explorer/. Data for the LED pathway[31] were obtained from the LED database: https://db1.ene.iiasa.ac.at/LEDDB/. Historical data on gross final energy from different energy carriers were taken from the online IEA Data and Statistics database, accessible at: https://www.iea.org/data-and-statistics. The data for energy system emissions, energy requirements for the energy system, net energy, and EROI generated in this study have been made available in the OSF online data repository: https://osf.io/v5nqg/?view_only=d28f6be45dc44dec884b0afa59098b76.

## Code availability

The Octave code used to calculate energy system emissions and energy for the energy system will be made available by the corresponding author upon reasonable request.

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

## Acknowledgements

We are grateful to Paul Brockway, Piers Forster, Eric Galbraith, Julia Steinberger, and Jeroen van den Bergh for their helpful comments. A.S. acknowledges financial support from la Caixa Foundation (ID 100010434, LCF/BQ/IN17/11620039). A.S. and G.K. also acknowledge support by the María de Maeztu Unit of Excellence (CEX2019-000940-M) grant from the Spanish Ministry of Science and Innovation.

## Author contributions

A.S., G.K., and D.W.O. all contributed to designing the project and writing up the results. A.S. led the project and the writing of the article, developed the model, collected and analysed the data, and produced the results, under the supervision of G.K. and D.W.O. G.K. had the initial idea for the project, discussed and refined the results with A.S., and contributed to the writing of the article. D.W.O. contributed to the analysis and discussion of the results, the preparation of visuals, and the writing of the article.

## Competing interests

The authors declare no competing interests.
