## [Peer review file · Nature Communications]

REVIEWER COMMENTS

Reviewer #1 (Remarks to the Author):

Energy requirements and carbon emissions for a low-carbon energy transition

General Comments

Overall, I found the article clearly written and well presented, with some interesting analysis applying EROIs (from a consumption-based LCA perspective) to the materialisation of energy system infrastructure in 1.5C pathways. However, I found the conclusions drawn to be misleading as they are based on a false premise that energy used in industrial output (for building energy infrastructure) and own-consumption of energy by energy infrastructure are not factored into 1.5C pathways interpreted by IAMs. My understanding of IAM sectoral energy accounting (including energy industry own-use, and upstream supply chain energy in resource extraction) is that this energy is accounted for, but not reported separately. So the authors' estimates are not additive (and so decrease 'net energy available to society') but rather add a nice interpretive overlay on the 1.5C pathways.

Specific Comments

p1-2

- I think the authors are referring to the embodied energy required for constructing energy-conversion infrastructure (like power plants or cars) but it's not always clear. The use of the term 'operate' and 'operational energy' I found confusing as operating a power plant (for example) implies combusting fossil fuels. I think the authors use operational energy to refer to what IEA energy balances refers to as energy industry 'own-consumption' or 'own use'.

- the authors assert that "the IPCC report's 1.5C pathways don't specify how much energy will be necessary to build and operate a low-carbon energy system" but this is misleading - the pathways implicitly capture embodied energy (e.g., through industrial energy use for producing cement, steel, cars, etc.) even it's not assigned to specific energy infrastructure.

- the ways IAMs account for operational (direct) and embodied (indirect) energy is certainly distinct from plant-specific EROI or LCA estimates, but this is a system boundary and model 'epistemology' issue, not a fundamental omission from IAMs ... EROI and LCA studies are analogous to consumption-based accounts, estimating direct and indirect effects from final investments in plants, factories or goods ... whereas IAMs are production-based accounts with industrial energy use and output flowing through into final investments. To say that the IAMs don't specify or make explicit how much goes into energy

infrastructure is right, but it's misleading to then infer that consequently they don't account for it (which is the tenor of the argument).

- for example the IAMs are typically calibrated to IEA energy balances which will assign energy used for concrete, steel, PV panels, etc. to industrial energy use. It's there, it's just not explicitly distinguished. IEA energy balances also distinguish own-consumption within the energy industry for plant operation (labelled as 'energy industry own use') as well as energy used in mining, resource transport, etc.

p3

- It seems misleading to apply a static logic that if constructing an energy system uses X energy then only Y energy would be left for "other societal uses" when total available energy is not a fixed quantity, and demand - supply interrelationships are dynamic, as are the pathways

p4

- The pathway nomenclature is a little confusing - it's not clear what the A, R, M suffixes denote, and it would be helpful to explain that S1 = SSP1-1.9W/m² etc (without the reader having to refer to SI to understand this).

p6

- In Fig1 legend, I have the same confusion as before with use of terms. Presumably by "energy system emissions" the authors are excluding emissions from the combustion of fossil fuels to generate power or produce oil products. Instead, they're using "energy system emissions" to refer just to own-consumption in energy facilities, upstream energy in extracting and transport, as well as embodied energy in construction. (After reading the main text, I then read the Methods, where it's made clear how the authors use the term).

p7

- I think the same EROI ranges per technology are the same for all pathways, but each pathway being analysed has a distinct narrative about technological change, global and regional convergence, land use dynamics, innovation in production, and so on, which would in turn affect how EROIs dynamically change. Under 1.5C pathways, much of the energy investment would be electrified (wherever possible) with associated efficiency gains. I couldn't work out from the Methods how fully this dynamic relationship between 1.5C storylines and future EROI estimates was accounted for. I could see a dynamic EROI was applied to wind and solar, but in the same way for all 1.5C pathways regardless of storylines. Then no dynamic EROIs were applied to other technologies due to lack of data. This makes it awkward at best to interpret the dynamic relationships between EROIs and the 1.5C pathways to which they are applied over an 80 year time horizon. A similar awkwardness applies to the dynamic estimation of carbon intensity per energy carriers which for electricity is from scenario data (so specific to the 1.5C

pathway being analysed) but which for other energy carriers is taken from the REMIND model (so not specific to the 1.5C pathway being analysed).

p8

- My interpretation of the main argument is that in 1.5C pathways, the embodied energy in construction of energy infrastructure and energy resource supply chains has ever-increasing importance for emissions. This is of course the case as 1.5C pathways almost by definition must have (1) very rapid decarbonisation of power production, and (2) electrification or other decarbonisation of other forms of secondary and final energy. Negative emissions are the wild card that allows (2) to relax. But the growing importance of energy required in supply chains (and energy industry own-consumption) is clearly understood under very constrained carbon budgets.

p9

- The implication drawn that “energy system emissions” (as defined by the authors) must be deducted from emission budgets shown in 1.5C pathways is based on the premise that these pathways do not already implicitly account for these emissions (in the production of steel, concrete, etc. as well as own-consumption in the energy industry). See above comment; I don’t think this is correct.

p10

- “a low-carbon energy transition would drive up the energy requirements of the energy system” ... I’m unclear from previous analysis whether this statement (and the first paragraph of this concluding discussion) is relative to a counterfactual future energy transition which is not low carbon, or whether it’s relative to today’s energy system which is smaller (in population and economic terms) than the future, so the statement is a simple scale effect

- the arguments about net energy to society I also found misleading as some of the 1.5C pathways (like LED and SSP1-1.9) by design reduce final energy through improvements in energy efficiency and energy service efficiency without reducing activity levels. The combination of renewables and end-use electrification in these pathways also leads to reductions in final energy as the overall energy-conversion chain from primary energy to useful service becomes more efficient. If energy is invested in renewables manufacture (and end-uses are electrified), then this energy investment effectively increases energy “available to society”. It’s not a zero-sum game in which energy invested in power plant infrastructure is deducted from a static total and so is not available for final use.

- so it seems like the key insight flows straight from the input assumptions on EROI: 1.5C pathways relying on fossil CCS, bioenergy, and BECCS have low EROIs so higher shares of embodied and own-consumption energy than 1.5C pathways relying on RE

p15

- Different version of same concern as before about what's already included vs. not included in 1.5C pathways, but here specifically in relation to BECCS (and fossil CCS): 1.5C pathways (and the IAMs used to characterise them) do pick up both emissions from land-use change and the energy penalties imposed by post-combustion CCS units. The assumptions different IAMs make are certainly open to critique - about how large the CCS energy penalties are, or about land-use change dynamics from bioenergy expansion. But it's misleading to imply that "energy system emissions" from BECCS are not factored into the 1.5C pathways. This is then stated baldly and misleadingly at the bottom of p15: "current pathways ... do not quantify the energy for the energy system." My interpretation of the authors' work is different - their consumption-based accounting lens helps understand the energy and emission shares of 1.5C pathways that are specifically for supplying, building, and running energy infrastructure - but this is colouring in detail not adding a large chunk of previously unaccounted for energy and emissions

Reviewer #2 (Remarks to the Author):

The paper investigates an interesting topic - the net-energy and emissions implied in transitioning to a low-carbon energy system. The authors have conducted a very extensive estimates of the energy system EROI which is significant contribution per se. As such the papers provides useful input the policy discussion surrounding the energy transition and the set-up of the energy system models (IAMs) to account for net energy and energy system emissions. Their approach is partially limited by the reliance on "optimal" IAM scenarios rather than a more flexible normative modeling approach. Nevertheless, within these constraints, their work deserves communication to the wider research community. It can be enhanced, focused and clarified after revisions. Detailed comments and suggestions follow:

1:

Ln 36-39: The sentence correctly claims that there is no discussion on material requirements. Nevertheless, it incorrectly claims that the referenced papers do not discuss energy requirements of the transition. In fact Ref. 3 (the Sower's Way paper) is specifically investigating this aspect (the net energy available during transition). This is corrected later - Ln 49-50

Suggest to remove Ref 3 from being cited in the Ln 36-39 and just cite in Ln 49-50.

Given that it is materially relevant to the topic, I recommend to normalize and summarize the estimated Energy Investment ratio of energy devoted to the energy system from Ref 3 and the other relevant publications (e.g. 12,18 etc) in an organized format - preferably a table - as a point for comparison.

2:

Paragraph on Ln 56:

The validity of using energy system EROI estimates given the difficulties and context specific assumptions for their calculation should be discussed. A good addition to this discussion is Palmer, G. & Floyd, J. An Exploration of Divergence in EPBT and EROI for Solar Photovoltaics. *Biophysical Econ Resour Qual* 2, 15 (2017).

3:

Ln. 88-90 - please turn the question into an affirmation. It is not asking the question but providing the answer - e.g. "we estimate how much energy will be available..."

4:

Ln 90-93 - The claim: "Ours is the first study to consider energy generation for the entire global energy system, including electricity, liquid fuels, gases, and solids including coal and biomass, and the first to consider dynamic changes in the energy requirements of different technologies." Does not seem supported. Most of the referenced studies earlier consider the first part of the sentence. As for the dynamic changes in the energy requirements of different technologies - other papers also considered learning curves in the EROI or CED of technologies.

The authors should provide stronger basis or more nuance for these claims.

5:

Ln 94 - Many IAMs can be significantly biased not only by discounting future catastrophic outcomes (1.Ackerman, F., DeCanio, S. J., Howarth, R. B. & Sheeran, K. Limitations of integrated assessment models of climate change. *Climatic Change* 95, 297–315 (2009).) but also towards replicating the status quo of technical options. For a detailed examination see Kaya, A., Csala, D. & Sgouridis, S. Constant elasticity of substitution functions for energy modeling in general equilibrium integrated assessment models: a critical review and recommendations. *Climatic Change* 43, 225–14 (2017). These aspects should at least be acknowledged/discussed since they form the basis of the analysis.

This is not just theoretical but has direct material implications to the findings of the paper. IAMs tendency to sustain status quo technologies would be biased towards expanding carbon capture and

storage (CCS) and biomass - since these are modeled as direct substitutes - which seems to be the case from the models presented. Since CCS and biomass generally have very low EROI, this biases the system net energy availability downwards. This is confirmed by the scenarios that emphasize variable renewables over biomass and CCS (see :Ln: 236-241)

6:

Ln: 113-116 and general EROI. The authors estimates of EROI for key transition technologies like PV at below 5 seems a severe underestimate of current status. For example, PV systems installed even in moderate to low insolation regions like Switzerland were shown to exceed 9 (1.Raugei, M. et al. Energy Return on Energy Invested (ERoEI) for photovoltaic solar systems in regions of moderate insolation: A comprehensive response. Energy Policy 102, 377–384 (2017).)

The authors seem to also significantly underestimate the effect of CCS on the EROI of fossil fuels or biomass options. Based on the methodology followed in Ref. 53, a reduction in the EROI by CCS of 30-40% should be expected and this is not observable in the Supplementary Fig. 1 and 2. In fact in the case of Gas with CCS or Biomass to Oil with CCS there is an increase!!! in the EROI compared to without CCS which is of course physically impossible (goes against 1st and 2nd law of thermodynamics as it provides additional work - the capture, compression and injection of the CO2 flue gas at no energetic cost). Please provide a detailed explanation on how the CCS EROI was calculated and crosscheck the analysis.

Given the uncertainty around EROI some kind of sensitivity analysis on this seems warranted. The authors are conducting this but the ranges of the EROI and the way this is estimated seems to be based on the assumption that high EROI is high across the board (Ln 591-595). This strikes this reviewer as a strange assumption, it is like trying to bracket average travel times assuming that all cars on the road are moving at their top speed - a highly unlikely situation. It would be better to segregate the effects of EROI by sector and focus on what is likely (that is high EROI RE) as opposed to what is unlikely (high EROI fossils that are known to be depleting and thus their EROI being have a downward trend tendency)

7:

Ln 142-143 - the use of the 4 + 10 scenarios is not clearly justified. It would be more informative and easier to follow if the scenarios were somehow categorized based on key properties. This can then be used in the Results/Discussion sections to support the narrative of the findings.

E.g. with a tabulated summary of the key results and their ranges. The table could be set-up by categorizing the 14 scenarios based on their common characteristics e.g. into scenarios with high biomass, high CCS, or high variable renewables and then provide numerically the range for each case of:

* System net energy

- * Energy system related emissions
- * Emissions available to non-energy activities
- * etc.

8:

Fig. 2 - it is difficult to reconcile the left (a,c,e) and right (b,d,f) panels - the left are supposed to be complete envelopes of all 14 scenarios yet the outlines are different than the right panels. This is explained due to "different EROI assumptions" in the description although the sub-title of the panels remains the same (i.e. high EROI for e, f).

Some additional explanation is needed on what the difference is and why it was introduced in the first place.

The explanation in the text seems Ln 185-188 seems contradicting and is also difficult to understand.

9:

Ln 215: what is meant by "jump" and "bump" in emissions? How is it different than the previous section where emissions are discussed as well? Is it the rate of emissions change that is discussed here? Or is it a change in the distribution of emissions (early vs. late?)

Once clarified, I recommend the title of the sub-section to be changed into a statement rather than a question.

10:

Overall net energy estimates are in line with expectations and previous work. Nevertheless, the statements on their evolution (e.g. in lines 259-260) on the drop of net energy per capita need some additional explanations. First, net energy per capita does not have to drop - it is a policy variable dependent on several rates (population growth, decarbonization rate, and RE adoption rate). The alternative approach of normative policy (as presented e.g. in the Sower's way - Ref. 3 or in the Ackerman et al. 2009 paper) avoids this type of outcome. It would be important to have this discussion somewhere - the scenarios are not the future but supposedly "optimal" paths to achieve decarbonization depending on policy choices.

11:

Ln 285-287. Another question set-up which could be better if it provided information/the answer.

12:

The decomposition analysis seems to provide marginal benefit (the conclusion in Ln 313-314 is rather trivial - more energy more emissions). The interesting part that should be investigated is on the marginal benefit of the change - is the relationship linear or non-linear? Can a substantial increase in final consumption be achieved with only a small increase in emissions? I expect it to be highly non-linear for high EROI high RE scenarios. The authors should investigate and report on this question instead. Perhaps they would need to examine this by different IAM scenarios and vary only this parameter.

Reviewer #3 (Remarks to the Author):

This is a thorough, thoughtful, and useful addition to the literature on the energy-emissions trap. The work is significant to the field, and notes further work required on the consumption side of the energy transition (i.e. net energy of replacing energy consuming devices). There will also be important considerations of the energy costs of maintaining a low-carbon energy system, and the remaining net energy available to society. Despite the need for further work, I find this methodology and the conclusions to be sound and ready for publication.

Response to Reviewers

Please find below our detailed responses to the comments from the three reviewers, which have improved the clarity and quality of our article. Reviewer comments are shown in *blue italics*, while our responses to these are in black. We did considerable changes to the manuscript – tracking them would impair readability. We have accordingly highlighted in **blue** only cases where new text/sentences were added. All line numbers that we use refer to the new manuscript.

Reviewer #1:

General Comments

Overall, I found the article clearly written and well presented, with some interesting analysis applying EROIs (from a consumption-based LCA perspective) to the materialisation of energy system infrastructure in 1.5C pathways.

Thank you for this positive assessment.

However, I found the conclusions drawn to be misleading as they are based on a false premise that energy used in industrial output (for building energy infrastructure) and own-consumption of energy by energy infrastructure are not factored into 1.5C pathways interpreted by IAMs. My understanding of IAM sectoral energy accounting (including energy industry own-use, and upstream supply chain energy in resource extraction) is that this energy is accounted for, but not reported separately. So the authors' estimates are not additive (and so decrease 'net energy available to society') but rather add a nice interpretive overlay on the 1.5C pathways."

The reviewer is correct that our estimates of energy requirements and emissions associated with the energy system are not additive. As the reviewer notes, “this energy is accounted for, but not reported separately”. We did not intend to claim that the energy requirements of the energy infrastructure are not accounted for in the pathways. They are indeed accounted for within the energy use of different sectors, but are not modelled/estimated explicitly within these sectors. This is important, because it means the IAM mitigation pathways do not provide us with information about how much of the total energy generation is required during the transition for the construction of low-carbon energy systems and the maintenance of energy supply. This in turn has implications for the energy (and associated emissions) left for other societal uses.

We have now rewritten the paragraph about the IAMs making clear that they do account for energy system energy and emissions, but that they do not report this separately (see lines 49-59). We have removed any ambiguous sentences in the article that could be interpreted as suggesting that IAMs do not account for energy inputs or emissions. Note though that when we talk about a decrease in net energy available to society, this is not in comparison to IAMs, but a decrease over time – a decrease that is accounted for in IAMs, but not acknowledged since IAMs do not report separately the amount of energy required for the energy system.

Specific Comments

p1-2

- I think the authors are referring to the embodied energy required for constructing energy-conversion infrastructure (like power plants or cars) but it's not always clear. The use of the term 'operate' and 'operational energy' I found confusing as operating a power plant (for example) implies combusting fossil fuels. I think the authors use operational energy to refer to what IEA energy balances refers to as energy industry 'own-consumption' or 'own use'.

Our terminology is consistent with the terminology used in the life-cycle assessment and net-energy/EROI literatures (see Arvesen et al., 2018; Castro and Capellan Perez, 2020; Sgouridis et al., 2019; and King and van den Bergh, 2018). However, we realise that we need to state clearly our definitions from the very beginning of the article to avoid misunderstandings. We now start the article with a definition of “energy for the energy system” and “energy system emissions” (lines 36-45), where we use also the language of the reviewer to make sure that our system boundaries are clear. Energy for the energy system includes the energy required for the construction (including decommissioning), operation, and maintenance of energy facilities like power plants and refineries as well as the energy required to produce and transport the energy carriers from the point of extraction to the end user. Therefore, our method includes the energy and emissions associated with the full life-cycle of energy facilities, upstream energy demand for extraction, processing and transportation of raw fuels, as well as the energy required for the delivery of energy carriers to the end user.

The difference between our terminology and the term “energy industry own use”, as defined in the IEA energy balances, is that energy industry own use accounts only for direct energy uses, but does not account for indirect energy uses (e.g. the energy used for transporting and distributing raw fuels and energy carriers), which are reported under final energy use. We explain this in lines 51-59, where we discuss how energy and energy system emissions are accounted for within IAMs.

- the authors assert that “the IPCC report’s 1.5C pathways don’t specify how much energy will be necessary to build and operate a low-carbon energy system” but this is misleading - the pathways implicitly capture embodied energy (e.g., through industrial energy use for producing cement, steel, cars, etc.) even it’s not assigned to specific energy infrastructure.

The reviewer is correct that the pathways capture the embodied energy, but the issue is that they do not model it explicitly and also do not report on it (hence our phrase that they do not “specify” it). As explained above, our point is not that this energy is not accounted for, but that the specific energy uses are not separated out in the pathways, which makes the energy requirements of the energy system during the transition difficult to quantify. This is an important blind spot and research gap as we explain in lines 60-63.

- the ways IAMs account for operational (direct) and embodied (indirect) energy is certainly distinct from plant-specific EROI or LCA estimates, but this is a system boundary and model ‘epistemology’ issue, not a fundamental omission from IAMs. EROI and LCA studies are analogous to consumption-based accounts, estimating direct and indirect effects from final investments in plants, factories or goods whereas IAMs are production-based accounts with industrial energy use and output flowing through into final investments. To say that the IAMs don’t specify or make explicit how much goes into energy infrastructure is right, but it’s misleading to then infer that consequently they don’t account for it (which is the tenor of the argument).

Yes, that’s correct and we thank the reviewer for making this distinction clear. We have incorporated the reviewer’s point in the new version of the manuscript (see lines 46-59). We

have also benefited from the reviewer's distinction between production-based and consumption-based accounting, which we now use to signal the difference of our approach (see lines 76-86).

- for example the IAMs are typically calibrated to IEA energy balances which will assign energy used for concrete, steel, PV panels, etc. to industrial energy use. It's there, it's just not explicitly distinguished. IEA energy balances also distinguish own-consumption within the energy industry for plant operation (labelled as 'energy industry own use') as well as energy used in mining, resource transport, etc.

We agree with the reviewer. In the revised article, we clarify that IAMs divide energy consumption and emissions into a few sectors, such as the industrial sector, but do not break down energy consumption within these sectors into particular economic activities (see lines 51-55). As such, the existing mitigation pathways in the IPCC literature do not allow us to calculate separately the energy and emissions associated with the transition, or know how much is effectively left for other uses after accounting for the needs of the energy system itself.

p3- It seems misleading to apply a static logic that if constructing an energy system uses X energy then only Y energy would be left for "other societal uses" when total available energy is not a fixed quantity, and demand - supply interrelationships are dynamic, as are the pathways

We did not intend to suggest that energy generation during a low-carbon energy transition is strictly constrained. We now make this clear in the section "Avoiding the energy trap" (lines 324-339). Our point is that substantial energy requirements for the energy system imply, other factors equal, less energy available for other end-uses, which should be made transparent. The "static approach" is used to illustrate the importance of the explicit modelling of energy requirements and the emissions associated with the energy system by integrating EROI analysis into IAM models. Moreover, one could argue that by integrating an EROI analysis into IAMs, these models would probably produce different energy and emissions pathways, with differences in the choice of energy generation technologies, as we argue in the discussion (lines 418-430).

A comparable approach distinguishing between the energy requirements of the energy system and the energy for "other societal uses" has been used previously in the EROI literature. See for example Sgouridis et al. (2016; <https://doi.org/10.1088/1748-9326/11/9/094009>) who argue that "The upfront energy invested in constructing a RE [i.e. renewable energy] infrastructure subtracts from the net energy available for societal energy needs, a fact typically neglected in energy projections". A similar analysis is conducted by King and van den Bergh at *Nature Energy* (2018; <https://doi.org/10.1038/s41560-018-0116-1>).

p4- The pathway nomenclature is a little confusing - it's not clear what the A, R, M suffixes denote, and it would be helpful to explain that SI = SSP1-1.9W/m2 etc (without the reader having to refer to SI to understand this).

We now provide an explanation of how we named the scenarios in the note below Table 1.

p6

- In Fig1 legend, I have the same confusion as before with use of terms. Presumably by "energy system emissions" the authors are excluding emissions from the combustion of fossil fuels to

generate power or produce oil products. Instead, they're using "energy system emissions" to refer just to own-consumption in energy facilities, upstream energy in extracting and transport, as well as embodied energy in construction. (After reading the main text, I then read the Methods, where it's made clear how the authors use the term).

Yes, that's correct. In the revised article, we start with a definition of "energy system emissions" to make things clearer from the beginning (see lines 36-45).

p7

- I think the same EROI ranges per technology are the same for all pathways, but each pathway being analysed has a distinct narrative about technological change, global and regional convergence, land use dynamics, innovation in production, and so on, which would in turn affect how EROIs dynamically change. Under 1.5C pathways, much of the energy investment would be electrified (wherever possible) with associated efficiency gains. I couldn't work out from the Methods how fully this dynamic relationship between 1.5C storylines and future EROI estimates was accounted for. I could see a dynamic EROI was applied to wind and solar, but in the same way for all 1.5C pathways regardless of storylines. Then no dynamic EROIs were applied to other technologies due to lack of data. This makes it awkward at best to interpret the dynamic relationships between EROIs and the 1.5C pathways to which they are applied over an 80 year time horizon..."

The reviewer is correct that our EROI scenarios are not tailored to the different narratives of the SSPs. Ideally our analysis would be designed as a model-specific and narrative-specific representation of EROI dynamics. Such an EROI framework could be integrated into the IAMs so it would be directly coupled to the modelling of energy and emissions in these mitigation pathways. However, this would require close collaboration with each of the six IAM modelling groups in charge of the SSP modelling experiments. Such an endeavour is, unfortunately, beyond the scope of our study. Instead, we have designed a generic EROI model, which overlays the energy requirements and energy system emissions on top of the existing scenarios. Our aim is to provide a range of estimates of energy system EROIs and indirect emissions in the transition, in order to demonstrate the relevance of net energy analysis being integrated into the IAM framework. Further research could provide more internally consistent extensions to existing IAMs. We now acknowledge the point made by the reviewer as an area for future research (see lines 420-425)

A similar awkwardness applies to the dynamic estimation of carbon intensity per energy carriers which for electricity is from scenario data (so specific to the 1.5C pathway being analysed) but which for other energy carriers is taken from the REMIND model (so not specific to the 1.5C pathway being analysed).

Our estimation of carbon intensity for carriers other than electricity is due to a lack of data from the emissions of these carriers in the mitigation pathways. The IAMC Scenario Explorer database only provides emissions from electricity generation but does not provide data for emissions from other carriers, so we could not use the carbon intensities of these carriers as they are implicitly assumed in some of the pathways. We calculated the carbon intensities in the pathways by referring to the emissions coefficients for secondary energy from the Emissions Database for Global Atmospheric Research (EDGAR), which is the approach used in the REMIND and IMAGE models. Our carbon intensity calculations are therefore fully in line with the pathways produced by these models. We could not do the same for AIM, WITCH, and GCAM because neither the specific emission coefficients, nor the original source of the

emission factors, are made publicly available. We could not estimate the carbon intensities of the energy carriers in the pathways produced by MESSAGE, as this model calculates the emissions at the primary energy stage, and not the secondary energy stage corresponding to the energy carriers. Therefore, in these cases, we used the approach from REMIND, which is a reasonable generalisation given data limitations.

p8

- My interpretation of the main argument is that in 1.5C pathways, the embodied energy in construction of energy infrastructure and energy resource supply chains has ever-increasing importance for emissions. This is of course the case as 1.5C pathways almost by definition must have (1) very rapid decarbonisation of power production, and (2) electrification or other decarbonisation of other forms of secondary and final energy. Negative emissions are the wild card that allows (2) to relax. But the growing importance of energy required in supply chains (and energy industry own-consumption) is clearly understood under very constrained carbon budgets.

The reviewer is correct in their interpretation of our argument. Note however that this is not our only finding – in the revised discussion section, we identify three core insights yielded by our analysis (see lines 384-416).

We are also not as confident as the reviewer is that the constraints posed by energy system emissions are clearly understood in the scenario literature (and possibly even less so in climate mitigation policy). The IAM mitigation pathways split the emissions into emissions from energy generation and direct consumption of energy carriers (e.g. in internal combustion engines). We illustrate the knowledge gap related to the aggregated reporting of emissions in the existing energy transition pathways by using the example of residual emissions from economic activities that are difficult to fully decarbonise, such as aviation, cement, and steel (see lines 240-239). Our comparison of energy system emissions with residual emissions shows that energy system emissions may take up most if not all of remaining emissions in some pathways. This result is not commonly known, as suggested by the reviewer. To our knowledge, it is not analysed or reported in the existing scenarios literature.

p9

- “The implication drawn that “energy system emissions” (as defined by the authors) must be deducted from emission budgets shown in 1.5C pathways is based on the premise that these pathways do not already implicitly account for these emissions (in the production of steel, concrete, etc. as well as own-consumption in the energy industry). See above comment; I don’t think this is correct”.

The reviewer is correct that energy system emissions are implicitly accounted for in the mitigation pathways. We do not claim that existing pathways do not account for these emissions, only that they do not model them explicitly. So the point is not that they should be “deducted”, but that we should have a clear view of how high they are, and what quantity of emissions (and energy) is left for other uses.

We also raise questions about the extent to which changing EROIs are captured by the production-based accounting of IAMs. Our estimates indeed suggest that some pathways may underestimate either residual emissions or energy system emissions, since our calculation of

energy system emissions under these pathways does not leave any room for residual emissions further down the century.

p10

- “a low-carbon energy transition would drive up the energy requirements of the energy system” ... I’m unclear from previous analysis whether this statement (and the first paragraph of this concluding discussion) is relative to a counterfactual future energy transition which is not low carbon, or whether it’s relative to today’s energy system which is smaller (in population and economic terms) than the future, so the statement is a simple scale effect”.

We now specify that we refer to the share of total energy generation that must be used for the construction, operation, and maintenance of the energy system if compared to the energy system of today (lines 280-283). Changes in the share of total energy going to the energy system do not necessarily depend on changes in scale.

- the arguments about net energy to society I also found misleading as some of the 1.5C pathways (like LED and SSP1-1.9) by design reduce final energy through improvements in energy efficiency and energy service efficiency without reducing activity levels. The combination of renewables and end-use electrification in these pathways also leads to reductions in final energy as the overall energy-conversion chain from primary energy to useful service becomes more efficient. If energy is invested in renewables manufacture (and end-uses are electrified), then this energy investment effectively increases energy “available to society”. It’s not a zero-sum game in which energy invested in power plant infrastructure is deducted from a static total and so is not available for final use.

We agree with the reviewer’s point that improved end-use efficiency in a low-carbon energy system would mean that the same quantity (or even more) energy services could be provided with less final energy. We have added this argument in the revised manuscript (lines 324-339).

- so it seems like the key insight flows straight from the input assumptions on EROI: 1.5C pathways relying on fossil CCS, bioenergy, and BECCS have low EROIs so higher shares of embodied and own-consumption energy than 1.5C pathways relying on RE

The assumptions on the EROI values are indeed important for the results. However, our EROI assumptions alone do not pre-define the outcome of our analysis. Our EROI values for all technologies are dynamic and depend on the deployment and utilisation of energy infrastructure (installed capacity) of different energy technologies, which are specific to each of the energy pathways (see the “Note on EROI dynamics in fossil fuels and biomass technologies” and the “Note on EROI dynamics of wind and solar power” in the Supplementary Information). Moreover, the energy requirements of the energy system depend on the specific model’s choices regarding the deployment of low-carbon technologies and energy losses in conversion and distribution. Furthermore, in calculating emissions associated with the transition, we link the energy requirements to the carbon intensities of different energy carriers, which are also endogenous to the mitigation pathways.

p15

- Different version of same concern as before about what’s already included vs. not included in 1.5C pathways, but here specifically in relation to BECCS (and fossil CCS): 1.5C pathways (and the IAMs used to characterise them) do pick up both emissions from land-use change and

the energy penalties imposed by post-combustion CCS units. The assumptions different IAMs make are certainly open to critique - about how large the CCS energy penalties are, or about land-use change dynamics from bioenergy expansion. But it's misleading to imply that "energy system emissions" from BECCS are not factored into the 1.5C pathways. This is then stated baldly and misleadingly at the bottom of p15: "current pathways ... do not quantify the energy for the energy system."

The characterisation of BECCS in the original version of the manuscript was indeed misleading. We have revised our argument, and now discuss our finding that energy system emissions are higher in the pathways that assume large-scale use of BECCS (lines 265-278). We also refer to concerns in the literature about whether existing mitigation pathways may underestimate emissions from BECCS (lines 403-406 and 770-779).

My interpretation of the authors' work is different - their consumption-based accounting lens helps understand the energy and emission shares of 1.5C pathways that are specifically for supplying, building, and running energy infrastructure - but this is colouring in detail not adding a large chunk of previously unaccounted for energy and emissions.

Even though the energy for the energy system and energy system emissions are implicitly accounted for in existing mitigation pathways, the importance of this energy and these emissions has not been assessed in the literature. We are therefore not merely colouring in details. There is currently a debate about whether a fast decarbonisation will reduce energy available for society or overshoot emissions. This is not something that can be answered by the existing IAM literature, and our model contributes novel findings to the debate. We find that mitigation pathways with faster and earlier decarbonisation have lower energy system emissions than pathways that delay action. We are not aware of a previous study that has modelled and robustly demonstrated this point.

In the new discussion section we further identify three main insights from our analysis (lines 384-416): we quantify for the first time the scale of emissions involved in ambitious low-carbon transitions; we identify a long-term problem of energy system emissions in pathways of slower decarbonisation and large-scale carbon removal; and we find a reduction in net energy available to society (although in some pathways this reduction may only take place in the short run).

We believe the above points make an important contribution to current understandings of ambitious mitigation and low-carbon transitions.

Once again, we thank the reviewer for the time they have put into reviewing our article. As the reviewer's inputs have greatly contributed to the clarity and scope of our article, in the eventuality of a publication, we would like to acknowledge these contributions explicitly.

Reviewer #2

The paper investigates an interesting topic - the net-energy and emissions implied in transitioning to a low-carbon energy system. The authors have conducted a very extensive estimates of the energy system EROI which is significant contribution per se. As such the papers provides useful input the policy discussion surrounding the energy transition and the set-up of the energy system models (IAMs) to account for net energy and energy system emissions. Their approach is partially limited by the reliance on “optimal” IAM scenarios rather than a more flexible normative modelling approach. Nevertheless, within these constraints, their work deserves communication to the wider research community. It can be enhanced, focused and clarified after revisions. Detailed comments and suggestions follow:

We thank the reviewer for these positive comments on our analysis, and recognition of its importance to the wider research community. We also appreciate the reviewer’s constructive suggestions, which have helped us to improve our method and to more clearly communicate our contribution.

1:

Ln 36-39: The sentence correctly claims that there is no discussion on material requirements. Nevertheless, it incorrectly claims that the referenced papers do not discuss energy requirements of the transition. In fact Ref. 3 (the Sower’s Way paper) is specifically investigating this aspect (the net energy available during transition). This is corrected later - Ln 49-50.

Thank you for this remark. We were citing the study to support the claim that IAMs, rather than the referenced papers, do not report on the material resources and energy needed to build a low-carbon energy infrastructure. We see that our placement of the reference could be interpreted differently, so we have removed it from this particular sentence, but cited the study at lines 60-62.

Given that it is materially relevant to the topic, I recommend to normalize and summarize the estimated Energy Investment ratio of energy devoted to the energy system from Ref 3 and the other relevant publications (e.g. 12,18 etc) in an organized format - preferably a table - as a point for comparison.

We now provide a table of the EROI of the total energy system from different studies in Supplementary Table 2. We compare the values from these studies to our own estimates in the “EROI estimates of different energy carriers and the overall energy system”, in the Supplementary Information.

2:

Paragraph on Ln 56:

The validity of using energy system EROI estimates given the difficulties and context specific assumptions for their calculation should be discussed. A good addition to this discussion is Palmer, G. & Floyd, J. An Exploration of Divergence in EPBT and EROI for Solar Photovoltaics. Biophysical Econ Resour Qual 2, 15 (2017).

Thank you for this suggestion. We have expanded the section on EROI studies in the introduction by pointing out the main weaknesses of the net energy literature and have deliberated on how to approach them in lines 138-145, citing the reference mentioned.

3:

Ln. 88-90 - please turn the question into an affirmation. It is not asking the question but providing the answer - e.g. "we estimate how much energy will be available..."

We have changed the question into an affirmation (lines 36-38).

4:

Ln 90-93 - The claim: "Ours is the first study to consider energy generation for the entire global energy system, including electricity, liquid fuels, gases, and solids including coal and biomass, and the first to consider dynamic changes in the energy requirements of different technologies." Does not seem supported. Most of the referenced studies earlier consider the first part of the sentence. As for the dynamic changes in the energy requirements of different technologies - other papers also considered learning curves in the EROI or CED of technologies.

The authors should provide stronger basis or more nuance for these claims.

The reviewer is correct that other studies have estimated the EROI pathways for different technologies and in different energy transitions. The novelty of our contribution is in the application of net energy analysis to the climate mitigation pathways from the IPCC literature. Previous applications of net energy analysis and life-cycle assessment have mostly been used in specific, usually conceptual, scenarios of energy transition but did not engage with the IAM mitigation pathways used by the IPCC. We now make this clear at lines 103-108.

5:

Ln 94 - Many IAMs can be significantly biased not only by discounting future catastrophic outcomes (Ackerman, F., DeCanio, S. J., Howarth, R. B. & Sheeran, K. Limitations of integrated assessment models of climate change. Climatic Change 95, 297–315 (2009).) but also towards replicating the status quo of technical options. For a detailed examination see Kaya, A., Csala, D. & Sgouridis, S. Constant elasticity of substitution functions for energy modelling in general equilibrium integrated assessment models: a critical review and recommendations. Climatic Change 43, 225–14 (2017). These aspects should at least be acknowledged/discussed since they form the basis of the analysis.

This is not just theoretical but has direct material implications to the findings of the paper. IAMs tendency to sustain status quo technologies would be biased towards expanding carbon capture and storage (CCS) and biomass - since these are modelled as direct substitutes - which seems to be the case from the models presented. Since CCS and biomass generally have very low EROI, this biases the system net energy availability downwards. This is confirmed by the scenarios that emphasize variable renewables over biomass and CCS (see :Ln: 236-241).

Thank you for this comment. We have incorporated the suggested argument and the references provided to highlight this point in the discussion section of the article (lines 431-439).

6:

Ln: 113-116 and general EROI. The authors estimates of EROI for key transition technologies like PV at below 5 seems a severe underestimate of current status. For example, PV systems installed even in moderate to low insolation regions like Switzerland were shown to exceed 9 (Raugei, M. et al. Energy Return on Energy Invested (ERoEI) for photovoltaic solar systems in regions of moderate insolation: A comprehensive response. Energy Policy 102, 377–384 (2017).)

The low EROI of PV in the years 2020-2030, as shown in Supplementary Fig. 1, is not due to our underestimation of EROI values, but due to an uneven balance between the upfront energy requirements and the energy generation of this technology during the 2020-2030 period. The EROI of PV in Supplementary Fig. 1 shows the ratio of energy generation and energy requirements for all the PV generation capacity that is installed during the respective period. A massive upscaling of PV systems results in substantial upfront energy investment, but most of the energy payback takes place after 2030.

We now clarify this point in lines 18-30 of the Supplementary Information.

The authors seem to also significantly underestimate the effect of CCS on the EROI of fossil fuels or biomass options. Based on the methodology followed in Ref. 53, a reduction in the EROI by CCS of 30-40% should be expected and this is not observable in the Supplementary Fig. 1 and 2. In fact in the case of Gas with CCS or Biomass to Oil with CCS there is an increase!!! in the EROI compared to without CCS which is of course physically impossible (goes against 1st and 2nd law of thermodynamics as it provides additional work - the capture, compression and injection of the CO2 flue gas at no energetic cost). Please provide a detailed explanation on how the CCS EROI was calculated and crosscheck the analysis.

Our estimates of technical EROI potential from CCS technologies are higher than the technical EROI potential from non-CCS technologies. Our estimates of energy penalty range from 22% to 30% and are therefore consistent with the values estimated in the reference cited by the reviewer.

The reason why the EROI of non-CCS technologies decreases below the EROI of CCS technologies in our analysis is because traditional fossil fuels, according to the SSP mitigation pathways data, become less utilised for energy generation later in the century (supposedly because they are less economical with a rising carbon price). With lower utilisation of fossil fuel technologies, the capacity factors of existing infrastructure decrease, which leads to a lower energy return and lower EROI. This result is the consequence of stranded fossil fuel investments in the future. Some power plants will work at reduced capacities, others will be closed down before the end of their lifetime. Thus the “real-world” EROI of these technologies will decline beyond the “real-world” EROI of CCS technologies, which will still operate further into the future, as their carbon intensity is lower in comparison.

We have provided a detailed explanation in the revised supplementary note (see lines 31-46 of the Supplementary Information).

Given the uncertainty around EROI some kind of sensitivity analysis on this seems warranted. The authors are conducting this but the ranges of the EROI and the way this is estimated seems to be based on the assumption that high EROI is high across the board (ln 591-595). This strikes this reviewer as a strange assumption, it is like trying to bracket average travel times assuming that all cars on the road are moving at their top speed - a highly unlikely situation. It would be better to segregate the effects of EROI by sector and focus on what is likely (that is

high EROI RE) as opposed to what is unlikely (high EROI fossils that are known to be depleting and thus their EROI being have a downward trend tendency).

We agree with the comments of the reviewer. Our EROI scenarios from the original version of the article covered the whole probabilistic range of possible EROIs, leading to some unlikely combinations.

The EROI literature broadly agrees about the likely decline in the EROI of fossil fuel technologies (at the primary energy stage), and that the EROI of renewables will increase. Such a trend is also consistent with the recent historical trend. To our knowledge there are no studies that would suggest the EROI of fossil fuels will increase. For bioenergy, we consider that EROI may decrease, as the expansion of bioenergy will push supply chains towards crops and areas with lower biomass yields and extend the distances of transportation.

Based on these assumptions, we have made a major change to the set of scenarios used in the analysis. The “high-EROI scenario” now describes a high-EROI trajectory for renewables and a median-EROI trajectory for bioenergy. In the “low-EROI scenario” we assume a median-EROI trajectory for renewables and a low-EROI trajectory for bioenergy. We assume the EROI of fossil fuels (at the primary energy stage) will decline from the present-day median EROI value, in line with the historical trend. To model the dynamic EROI of fossil fuels, we have used the equation of exponential decay from Daly et al. (2011) and Sgouridis et al. (2016), and calibrated the time decay coefficients using the data from Brockway et al. (2019).

The changes are reflected throughout the article (see in particular lines 146-163, 569-577, and Fig. 2). We have also added new tables and figures to the Supplementary Information, which provide a detailed overview of our assumptions for different EROI scenarios (see Supplementary Tables 2-7 and Supplementary Figs. 3-4).

7:

Ln 142-143 - the use of the 4 + 10 scenarios is not clearly justified. It would be more informative and easier to follow if the scenarios were somehow categorized based on key properties. This can then be used in the Results/Discussion sections to support the narrative of the findings.

E.g. with a tabulated summary of the key results and their ranges. The table could be set-up by categorizing the 14 scenarios based on their common characteristics e.g. into scenarios with high biomass, high CCS, or high variable renewables and then provide numerically the range for each case of:

** System net energy*

** Energy system related emissions*

** Emissions available to non-energy activities*

** etc....*

The four mitigation pathways selected are the IPCC’s archetypal scenarios. They represent conceptually different energy and emissions pathways for limiting global warming to 1.5 degrees. These pathways are distinct with regards to energy consumption, the deployment of negative emissions, and different energy generation technologies. The intent of Table 1 is to

summarise the key characteristics of the archetypal pathways, before we provide the results on net energy and energy system emissions.

We have selected the ten additional pathways in order to increase the sample of transitions used in the analysis. However, the bulk of our comparative analysis is focused on the four distinct archetypal pathways, which serve the purpose identified by the reviewer here.

In response to the reviewer's comment, we have extended Table 1 with information on the different technology shares in total energy generation. We have also tried to better explain our choice of the four illustrative pathways (lines 176-191).

8:

Fig. 2 - it is difficult to reconcile the left (a,c,e) and right (b,d,f) panels - the left are supposed to be complete envelopes of all 14 scenarios yet the outlines are different than the right panels. This is explained due to "different EROI assumptions" in the description although the sub-title of the panels remains the same (i.e. high EROI for e, f).

Some additional explanation is needed on what the difference is and why it was introduced in the first place.

The explanation in the text seems Ln 185-188 seems contradicting and is also difficult to understand...

We have changed Fig. 2 by removing the panels on the right. We now plot the envelopes of the three EROI scenarios, which cover the whole range of the 14 pathways used in our analysis, with the four illustrative pathways displayed on top of these envelopes.

9: Ln 215: what is meant by "jump" and "bump" in emissions? How is it different than the previous section where emissions are discussed as well? Is it the rate of emissions change that is discussed here? Or is it a change in the distribution of emissions (early vs. late?)

Once clarified, I recommend the title of the sub-section to be changed into a statement rather than a question.

We have removed the reference to the "bump" and now consistently refer to a "jump" in emissions. We have further clarified that the jump refers to the increase in annual energy system emissions, not the share of total emissions that energy system emissions represent. We also clarify that the jump is mostly due to the surging upfront energy requirements for building a low-carbon energy system during the initial push for the energy transition (lines 250-253 and 260-262). We have also changed the heading into a statement, as suggested by the reviewer.

10:

Overall net energy estimates are in line with expectations and previous work. Nevertheless, the statements on their evolution (e.g. in lines 259-260) on the drop of net energy per capita need some additional explanations. First, net energy per capita does not have to drop - it is a policy variable dependent on several rates (population growth, decarbonization rate, and RE adoption rate). The alternative approach of normative policy (as presented e.g. in the Sower's way - Ref. 3 or in the Ackerman et al. 2009 paper) avoids this type of outcome. It would be important to have this discussion somewhere - the scenarios are not the future but supposedly "optimal" paths to achieve decarbonization depending on policy choices.

We appreciate this comment, as it shows that further elaboration on the implications of lower net energy is required. In response, we have added a new paragraph where we explain that lower net energy is not due to biophysical constraints to energy generation, but due to IAM assumptions on the cost-effectiveness of energy efficiency improvements. We also refer to the argument from the literature on energy sufficiency (i.e. that lower net energy does not necessarily imply a lower access to energy services; see lines 324-329). In relation to the decline in net energy, we discuss the feasibility of a hypothetical mitigation pathway that would break the link between energy consumption and energy system emissions (see lines 359-366). Finally, we use the suggested references from the reviewer to deliberate on the limitations of IAMs (lines 431-439).

11:

Ln 285-287. Another question set-up which could be better if it provided information/the answer.

We have changed the question into a statement/description of our approach (lines 342-345).

12:

The decomposition analysis seems to provide marginal benefit (the conclusion in ln 313-314 is rather trivial - more energy more emissions). The interesting part that should be investigated is on the marginal benefit of the change - is the relationship linear or non-linear? Can a substantial increase in final consumption be achieved with only a small increase in emissions? I expect it to be highly non-linear for high EROI high RE scenarios. The authors should investigate and report on this question instead. Perhaps they would need to examine this by different IAM scenarios and vary only this parameter.

We conducted a multiple regression panel data analysis to explore which factors vary most between the pathways in terms of their effects on energy-system emissions. As we now make clear, the finding is not only that “more energy results in more energy system emissions”. We find that both final energy use and the choice of technologies have a large effect, while changes in EROI have a weaker effect, particularly later on in the transition (see lines 342-350 and 351-374).

That said, we found the reviewer’s comment very helpful, as it directed us to better investigate the conditions that can limit or amplify energy system emissions. In the revised manuscript, we have changed the approach in our analysis. Instead of including the decomposition analysis, which provided limited additional explanation of the underlying emissions factors, we now analyse the relationships between the main causal factors of energy system emissions. We explore how the importance of different factors for energy system emissions changes over time. We show that energy use is a major factor in emissions during the initial push for the transition, but that the phase-out of conventional fossil fuels without CCS becomes the dominating factor later on in the transition. Mitigation pathways that prioritise the development of renewables and nuclear energy have lower energy system emissions than pathways that continue using fossil fuels and compensate for their emissions using BECCS. Finally, we discuss a hypothetical mitigation pathway that could break the link between energy and emissions (see lines 359-366).

Reviewer #3:

This is a thorough, thoughtful, and useful addition to the literature on the energy-emissions trap. The work is significant to the field, and notes further work required on the consumption side of the energy transition (i.e. net energy of replacing energy consuming devices). There will also be important considerations of the energy costs of maintaining a low-carbon energy system, and the remaining net energy available to society. Despite the need for further work, I find this methodology and the conclusions to be sound and ready for publication.

Thank you very much for your recognition of our contribution. In the new version of the article, we have extended our analysis of the energy-emissions trap by investigating the specific conditions under which energy system emissions could be reduced and deliberated on a hypothetical mitigation pathway where this could be achieved without reducing energy consumption. We have also pointed out the underlying reasons why the existing mitigation pathways typically diverge from a pathway of low energy system emissions.

REVIEWER COMMENTS

Reviewer #1 (Remarks to the Author):

Energy requirements and carbon emissions for a low-carbon energy transition [REVISED]

I appreciate the authors' evident efforts to address my concerns expressed in the first round of reviews, but I remain concerned that the essential argument and framing of the paper remains unchanged, and that using a consumption-based lens to analyse production-based energy and emission accounts is valid as a post-processing or interpretive step for IAM results but is problematic as currently applied in an apples to oranges methodological comparison.

Three illustrative issues:

- (1) operational vs embodied energy;
- (2) net energy available to society;
- (3) consumption-based vs production-based methodologies.

The language used in the revised draft is still confusing in the lack of clear separation between the energy required to operate energy infrastructure (e.g., gas used to run a power plant) and the energy required to build energy infrastructure (e.g., coal used in a blast furnace to make steel used to construct a power plant). Until operational energy is clearly distinguished from embodied energy, the language used will be confusing to the energy systems and IAM community who may be unfamiliar with norms in the EROI community.

A clear distinction between operational and embodied energy is important for understanding the results and arguments. It is self-evident that operating a low carbon energy system requires large amounts of energy as converting primary energy resources into useful energy forms is the energy system's fundamental purpose. If these primary energy resources are renewable, then emissions are low; vice versa if these primary energy resources are non-renewable. However I think the authors' argument is that constructing and maintaining the energy system requires large amounts of energy as the capital intensive shift from fossil fuels to renewables, nuclear, and bioenergy in 1.5C pathways implies an increased upfront investment of energy in materialising the new infrastructure. But the way the article is worded makes this unclear. As the authors make clear in their response to reviewers, one of their motivations for their analysis is that IAMs don't report separately embodied energy in the construction

of new energy infrastructure. (The same applies to transport infrastructure, buildings infrastructure, and so on). However, the IAMs do report primary energy resources going into the operation of the global energy system (as well as energy industry own use). So the contribution of the article requires a clear separation of operational and embodied energy and a focus on the latter. Currently these are blended into the tautological “energy for the energy system” which combines construction, operation, maintenance, transport (or ‘well to wheel’ in energy in transport parlance). The definition added in lines 36-45 is clear until the last sentence in which “operation” shifts to own consumption, which presumably is in line with energy industry own use (IEA energy balances) as opposed to “energy required to produce ... energy carriers”. As all 1.5C pathways see high levels of electrification and production of low-carbon liquids or gases (e.g., H₂ from electrolysis), the energy required to produce energy carriers is obviously large, and the subject of countless IAM studies. The confusion continues in the next paragraph which states that IAMs do not report “the energy needed to build, operate and manage the energy infrastructure”. They do report operational energy in the sense of producing energy carriers, and generally they report energy industry own use (the other use of “operational” which the authors use). They do not separate out energy needed to build and manage energy infrastructure from their industry and transport sectoral reporting, true. This is the piece of the puzzle I think the authors should focus on, rather than blending it with operational energy which is an IAM mainstay not a “blind spot”.

The authors still seem to treat “net energy available to society” as a fixed quantity against which energy system energy is deducted. Cumulative emissions for a given global warming outcome is a fixed quantity (albeit an uncertain one) but total energy is not. As the authors’ estimates of “energy for the energy system” include operational energy to produce energy carriers (e.g., gas to generate electricity, or renewables to produce hydrogen), then this energy in its useful forms is self-evidently available to society. Indeed, converting primary energy into useful forms available to society is the function of the global energy system. (See point above). For the embodied energy in constructing power plants or the transport energy in moving energy carriers, I can see that this reduces the useful energy available for end-users, but to say that this is somehow not available to society is misleading. It’s not available directly as useful energy, but it is available indirectly as embodied energy in capital formation. The IAMs account for this, albeit from their production-based and sectoral standpoint (i.e., it’s picked up in industrial energy use or in transport energy use). IAMs also report final energy in quantities available to end-users (or in the authors’ language “available to society”) with these intermediate or upstream uses of energy netted off. As an additional, related concern, I don’t follow the “reduced energy availability” argument as total primary energy resources converted into useful energy or capital formation by the global energy system is not fixed - only emissions are (to stay within 1.5C carbon budgets). So the analysis imposes a static view on a dynamic problem, and seemingly doesn’t allow for more energy infrastructure that requires more energy invested in its materialisation (even if less energy required to run it) as a means of ensuring more useful energy is delivered as electricity, heat, liquid fuels etc. to final users in households and firms. Again, to be clear, this is an energy argument, not an emissions argument ... but the authors bundle these together. For example, lines 109-110: “Our study separates between the energy and emissions associated with the energy system, and the energy and emissions left for other societal uses.” As a final observation, as the authors’ analysis is based on EROI and as the 1.5C pathways analysed have defined cumulative emissions (defined by the warming outcome) and as arguments about BECCS versus renewables are well worn, I think the more useful insights are on the energy implications

of low-carbon infrastructure transitions not the emission implications (which are constrained across all pathways, with negative emissions as a valve used not at all in LED but a lot in SSP5-1.9W/m2). Some of the implications for emissions are duplicative of core IAM competences, and/or are self-evident as in the statement in the response to reviewers that: “We find that mitigation pathways with faster and earlier decarbonisation have lower energy system emissions than pathways that delay action.” As decarbonisation means reducing energy system emissions, this is also tautological.

Both these concerns, and other more minor ones (see previous round of review comments inc. static EROIs vs dynamic pathways, and generic assumptions across very different 1.5C pathways) are the result of how the paper is framed as an IAM critique, which leads naturally to the recommendation that “energy requirements and emissions should be explicitly modelled in the next generation of low-carbon mitigation pathways”. I think this is misguided as it would - following the authors - imply a consumption-based overlay on the production-based energy and emissions frameworks used in IAMs. These are different modelling epistemologies, awkward if not impossible to simply combine. I think the consumption-based EROI aspect of the authors’ work is of interest, but this is a compare-and-contrast exercise with what IAMs report (at least in terms of upstream energy and emissions from the energy sector), not an embedding exercise of EROIs within IAMs as the authors argue. So the key section to frame the paper is really the one headed “Estimating energy requirements and emissions” rather than the preceding one, noting that the usefulness of the EROI approach in this context is its ability to estimate energy for construction, maintenance, and decommissioning (as opposed to operation which IAMs already report). Taking data from 1.5C pathways makes the EROI analysis a useful post-processing step for IAM results ... but also means that the operational energy part of the EROI calculations should be stripped out and compared with the IAM reporting for the upstream energy sector.

Reviewer #2 (Remarks to the Author):

The authors have satisfactorily addressed the comments of the reviewers. The paper provides a substantial contribution to the field and this reviewer recommends it for publication in its current form.

Response to Reviewers

Please find below our detailed responses to the comments of the two reviewers. Reviewer comments are shown in *blue italics*, while our responses are in black. We have highlighted in blue the parts of the manuscript where changes or new sentences were added. We made no changes to the supplementary information. All line numbers that we use refer to the new manuscript.

Reviewer #1:

I appreciate the authors' evident efforts to address my concerns expressed in the first round of reviews, but I remain concerned that the essential argument and framing of the paper remains unchanged, and that using a consumption-based lens to analyse production-based energy and emission accounts is valid as a post-processing or interpretive step for IAM results but is problematic as currently applied in an apples to oranges methodological comparison.

Three illustrative issues:

- (1) operational vs embodied energy;*
- (2) net energy available to society;*
- (3) consumption-based vs production-based methodologies.*

Thank you for recognising our efforts to address the concerns raised.

In the new revised version, we have changed the framing by removing any reference to IAMs at the opening of the article. Our contribution stands independently of IAMs. Our article is the first EROI-based calculation of energy requirements and emissions associated with the global energy system in a low-carbon transition. We did not aspire to a methodological comparison between IAMs and EROI, and we recognize now that by opening the article with a critique of IAMs the reader may get distracted from our main message, and expect from this article such a comparison and a more in-depth critique of the IAMs. That was not our intention, or the main point of our article.

We address the three illustrative issues in a point-by-point response below.

The language used in the revised draft is still confusing in the lack of clear separation between the energy required to operate energy infrastructure (e.g., gas used to run a power plant) and the energy required to build energy infrastructure (e.g., coal used in a blast furnace to make steel used to construct a power plant). Until operational energy is clearly distinguished from embodied energy, the language used will be confusing to the energy systems and IAM community who may be unfamiliar with norms in the EROI community.

We have changed our definition of the energy requirements, and now clearly distinguish between the energy required to operate energy infrastructure and the energy required to build infrastructure (see lines 481-488).

We hope this set of definitions offers clarity. Please note also that Nature journals have in the past published other EROI studies using the comparable approaches to ours (Brockway et al., 2019; King and Van Den Bergh, 2018; Sgouridis et al., 2019), and that the norms used by the EROI community are well established in the field.

A clear distinction between operational and embodied energy is important for understanding the results and arguments. It is self-evident that operating a low carbon energy system requires large amounts of energy as converting primary energy resources into useful energy forms is the energy system's fundamental purpose. If these primary energy resources are renewable, then emissions are low; vice versa if these primary energy resources are non-renewable. However I think the authors' argument is that constructing and maintaining the energy system requires large amounts of energy as the capital intensive shift from fossil fuels to renewables, nuclear, and bioenergy in 1.5C pathways implies an increased upfront investment of energy in materialising the new infrastructure.

Indeed, there are important differences in energy requirements between different energy technologies, which we describe in lines 509-514 and lines 611-619. For variable renewables (like wind and photovoltaics), almost all the energy requirements are for the upfront energy for the construction of energy infrastructure, whereas for fossil fuels with CCS and bioenergy, most of the energy requirements are for operation and maintenance.

We reflect on the importance of technological choices in the section “Avoiding the energy trap”, where we report on the increasing share of energy for the energy system due to the upfront energy requirements of low-carbon energy infrastructure, during the initial push for the transition (lines 266-269). We report the contrasting long-term increase in the share of energy for the energy system in pathways that prefer fossil fuels and bioenergy with carbon capture and storage over renewables in lines 293-297. The importance of different technological choices is illustrated with the overall EROI trajectories and the share of energy for the energy system in different pathways in Figure 4.

But the way the article is worded makes this unclear. As the authors make clear in their response to reviewers, one of their motivations for their analysis is that IAMs don't report separately embodied energy in the construction of new energy infrastructure. (The same applies to transport infrastructure, buildings infrastructure, and so on). However, the IAMs do report primary energy resources going into the operation of the global energy system (as well as energy industry own use).

Our motivation was to calculate the energy and emissions associated with the transformation of the energy system, which has not been calculated before, and that is not directly retrievable from IAMs. Our motivation, however, was not to compare our approach to IAMs, or to delve deeper into what portions of energy requirements and emissions for the energy system IAMs account for. This is a rather separate research question that we now introduce at the end of the

article, where we discuss future research directions. We no longer motivate, or open the article, with the reference to IAMs, to avoid a distraction towards such methodological comparisons, which are not the objective of this article.

For the sake of argument, let us note that IAMs indeed report on the supply of primary energy resources, and on the energy industry own use. The distinction between our boundaries of analysis and the boundaries of the IAMs is that our boundary for the energy required for operation and maintenance extends beyond the IEA's definition of energy industry own use. For liquid fuels and gas, our boundary includes inputs for transporting energy carriers, which increases the energy requirements for operation and maintenance, if compared to the boundary at the energy industry own use. For electricity generation from renewables, our boundary includes the energy requirements for the maintenance of renewable infrastructure (e.g. the energy required to service wind farms).

Our definition of boundaries is consistent with the standard boundaries in the LCA literature see (Arvesen et al., 2018; Pehl et al., 2017) and in the EROI literature (Brockway et al., 2019; Castro and Capellán-Pérez, 2020) and is different from the boundaries of analysis in IAMs.

So the contribution of the article requires a clear separation of operational and embodied energy and a focus on the latter. Currently these are blended into the tautological “energy for the energy system” which combines construction, operation, maintenance, transport (or ‘well to wheel’ in energy in transport parlance). The definition added in lines 36-45 is clear until the last sentence in which “operation” shifts to own consumption, which presumably is in line with energy industry own use (IEA energy balances) as opposed to “energy required to produce ... energy carriers”. As all 1.5C pathways see high levels of electrification and production of low-carbon liquids or gases (e.g., H₂ from electrolysis), the energy required to produce energy carriers is obviously large, and the subject of countless IAM studies.

We now clearly separate between the “operational and embodied energy”, or as we call it, the energy required for operation and maintenance, and the energy required for construction. We hope this provides sufficient definitional separation. After the above comment of the reviewer, we realized that there was a language issue in our definition, which created confusion. We had written that *energy for the energy system includes... the energy required to produce ... the energy carriers...*. By that we meant only energy inputs that are required to manage the energy infrastructure and the process of energy conversion (including energy required to transport energy carriers), NOT the energy embodied in primary resources/fuels which are converted into useful carriers. We have therefore revised our definition to avoid further misunderstandings (see lines 40-44) and extended the description of energy inputs that we account for in the methods (see lines 481-488).

However, we do not agree with the reviewer that the contribution of the article requires a more thorough deliberation on the importance of “operational and embodied energy”. In our method to calculate the energy requirements and emissions, we indeed differentiate between different energy requirements, but we do not report on different emissions from energy requirements for construction, operation and maintenance because this is not of primary importance for the article.

The contribution of the article is to calculate the total, which has never been calculated before. There might be an argument that follow-up work should delve further into the contribution of the energy needs of different processes (construction, operation, maintenance, decommissioning), and how they might differ between technologies or scenarios. As we stated in the reply to the previous comment, we do point out the importance of upfront energy costs for construction of renewables vs. the operational costs of fossil fuels and bioenergy technologies. Our method indeed provides the tools and data for such analysis. But this level of descriptive detail would only distract from what is in our opinion the core contribution of the article (i.e. to derive the grand total, and see how it evolves over time under different assumptions and in different scenarios).

The confusion continues in the next paragraph which states that IAMs do not report “the energy needed to build, operate and manage the energy infrastructure”. They do report operational energy in the sense of producing energy carriers, and generally they report energy industry own use (the other use of “operational” which the authors use). They do not separate out energy needed to build and manage energy infrastructure from their industry and transport sectoral reporting, true. This is the piece of the puzzle I think the authors should focus on, rather than blending it with operational energy which is an IAM mainstay not a “blind spot”.

We have deleted this section of text, and reframed the article to avoid any confusion. Our goal is not to calculate only what is not reported by IAMs. We are after the grand total.

The authors still seem to treat “net energy available to society” as a fixed quantity against which energy system energy is deducted. Cumulative emissions for a given global warming outcome is a fixed quantity (albeit an uncertain one) but total energy is not.

Total energy is not a fixed quantity in general, but it is given by each of the 1.5 C scenarios examined here. Our conclusions about the decline in net energy are derived from the projected energy pathways in the mitigation scenarios, which all assume that growth in final energy will decline at some point during the transition, or become negative. Whether these pathways adequately represent the full range of possible energy futures is beyond the scope of our analysis.

As the authors’ estimates of “energy for the energy system” include operational energy to produce energy carriers (e.g., gas to generate electricity, or renewables to produce hydrogen), then this energy in its useful forms is self-evidently available to society.

This is a misunderstanding of our definition of the energy required for operation and maintenance from the previous version of the article, which we have addressed with an improved definition in lines 40-44. As we now describe in the methods, the energy required for operation and maintenance does not include the raw energy content of energy resources/fuels (e.g. heating value of gas to generate electricity) that are converted into useful carriers. It only accounts for energy inputs that are required to procure and process the resources/fuels, operate the energy infrastructure, and deliver the carriers to the end user (see added explanation in the methods, lines 493-497).

Indeed, converting primary energy into useful forms available to society is the function of the global energy system. (See point above). For the embodied energy in constructing power plants or the transport energy in moving energy carriers, I can see that this reduces the useful energy available for end-users, but to say that this is somehow not available to society is misleading. It's not available directly as useful energy, but it is available indirectly as embodied energy in capital formation. The IAMs account for this, albeit from their production-based and sectoral standpoint (i.e., it's picked up in industrial energy use or in transport energy use). IAMs also report final energy in quantities available to end-users (or in the authors' language "available to society") with these intermediate or upstream uses of energy netted off.

The framing of the reviewer is compatible with our framing. Our differences here originate from different knowledge traditions. The term 'net energy' originates from the field of energy economics, and there are several publications in Nature that use it in the way we use it here. Net energy is the part of total energy generation that can be dedicated to provide for society's needs (i.e. in contrast to the energy requirements of the energy system). In the energy economics literature the term "net energy available to society", which we use, is well established (see its use in (Brockway et al., 2019; Carbajales-Dale et al., 2014), as examples). The energy that is invested into building energy infrastructure (capital) to produce energy is not available to society for other productive uses, so it is subtracted from net energy.

Note also that there is a difference between the definition of final energy and net energy in our approach. Net energy deducts from the total the final energy inputs that go into the build-up and maintenance of energy infrastructure and transportation of final energy carriers. This should be clear in the updated definition of net energy (lines 104-106) and by the definition of the boundaries in our manuscript, which are illustrated in Supplementary Figure 7.

As an additional, related concern, I don't follow the "reduced energy availability" argument as total primary energy resources converted into useful energy or capital formation by the global energy system is not fixed - only emissions are (to stay within 1.5C carbon budgets). So the analysis imposes a static view on a dynamic problem, and seemingly doesn't allow for more energy infrastructure that requires more energy invested in its materialisation (even if less energy required to run it) as a means of ensuring more useful energy is delivered as electricity, heat, liquid fuels etc. to final users in households and firms. Again, to be clear, this is an energy argument, not an emissions argument ... but the authors bundle these together. For example, lines 109-110: "Our study separates between the energy and emissions associated with the energy system, and the energy and emissions left for other societal uses."

Here, our analysis is motivated by extensive literature that expresses concerns about the decline in net energy available to society, as is briefly summarised in lines 57-63. However, we do not suggest that energy during a low-carbon energy transition is biophysically constrained as the carbon budget is. We make this clear in the article in lines 303-308:

“All pathways suggest an inevitable decline in per capita net energy at some point during the transition. However, this finding does not mean that energy scarcity is an unavoidable feature of any low-carbon energy transition.”

Our conclusions about the decline in net energy are derived from the projected energy pathways in the mitigation scenarios, which all assume that growth in energy declines at some point during the transition, or becomes negative. Whether these pathways adequately represent the full range of possible energy futures is beyond the scope of our analysis.

As a final observation, as the authors’ analysis is based on EROI and as the 1.5C pathways analysed have defined cumulative emissions (defined by the warming outcome) and as arguments about BECCS versus renewables are well worn, I think the more useful insights are on the energy implications of low-carbon infrastructure transitions not the emission implications (which are constrained across all pathways, with negative emissions as a valve used not at all in LED but a lot in SSP5-1.9W/m2). Some of the implications for emissions are duplicative of core IAM competences, and/or are self-evident as in the statement in the response to reviewers that: “We find that mitigation pathways with faster and earlier decarbonisation have lower energy system emissions than pathways that delay action.” As decarbonisation means reducing energy system emissions, this is also tautological.

We agree that the net energy implications of a low-carbon energy transition are important, which is why we extensively report on them in the energy trap section of the article. However, we continue to believe that our main contribution is the calculation of the cumulative emissions associated with the transition.

As we have noted above, the total energy system emissions have not been calculated before. It is important to know the magnitude of these emissions, and the challenges (if any) such emissions pose to mitigation pathways. Beyond that, we also ask what do these energy system emissions imply for the end-user? What is the remaining carbon budget from fossil fuel energy carriers that can be used on the consumption side of the economy (i.e. energy embodied in electricity and fuels used by end consumers), after we account for the emissions that are effectively bootstrapped to the transformation of the energy system? We now better elaborate on this framing in the conclusion in lines 392-400.

Both these concerns, and other more minor ones (see previous round of review comments inc. static EROIs vs dynamic pathways, and generic assumptions across very different 1.5C pathways) are the result of how the paper is framed as an IAM critique, which leads naturally to the recommendation that “energy requirements and emissions should be explicitly modelled in the next generation of low-carbon mitigation pathways”. I think this is misguided as it would - following the authors - imply a consumption-based overlay on the production-based energy and emissions frameworks used in IAMs. These are different modelling epistemologies, awkward if not impossible to simply combine. I think the consumption-based EROI aspect of the authors’ work is of interest, but this is a compare-and-contrast exercise with what IAMs report (at least in terms of upstream energy and

emissions from the energy sector), not an embedding exercise of EROIs within IAMs as the authors argue.

We recognise that previous versions of this article contained a critique of IAMs. In response to the reviewer's concerns, we have removed this text, and now only refer to IAMs at the end, as part of a possible future research agenda (see lines 401-408 and 416-425).

Note that we do not agree that our approach is impossible to combine with existing IAMs. Some attempts in this direction have been made, for example (Pehl et al., 2017), have used a similar framing of the indirect emissions from energy technologies, and have extended the IAM REMIND model with a life-cycle analysis. Other examples are the MEDEAS (Capellán-Pérez et al., 2019) and EUROCLIM (D'Alessandro et al., 2020) models, which incorporate EROI into their modelling of the energy system.

So the key section to frame the paper is really the one headed "Estimating energy requirements and emissions" rather than the preceding one, noting that the usefulness of the EROI approach in this context is its ability to estimate energy for construction, maintenance, and decommissioning (as opposed to operation which IAMs already report). Taking data from 1.5C pathways makes the EROI analysis a useful post-processing step for IAM results ... but also means that the operational energy part of the EROI calculations should be stripped out and compared with the IAM reporting for the upstream energy sector.

We agree that the core contribution of our article is the calculation of energy requirements and emissions associated with the energy transition, which we have calculated using a novel EROI approach. As we have explained, our goal is not to compare the EROI methodology/epistemology of energy system modelling with the IAM methodology/epistemology, but to provide a complementary framing of energy system boundaries, which we found necessary for estimating the (indirect) energy requirements and emissions associated with the energy system during the transition. Our work complements the extensive IAM literature on mitigation scenarios, and does not challenge their important contribution.

Reviewer #2 (Remarks to the Author):

The authors have satisfactorily addressed the comments of the reviewers. The paper provides a substantial contribution to the field and this reviewer recommends it for publication in its current form.

Thank you very much for your recognition of the contribution made by the article, and for the suggestions in the previous round of reviews. Your comments have improved our discussion of the relevant literature in the field, and have assisted us in framing the contribution more clearly.

References

- Arvesen, A., Luderer, G., Pehl, M., Bodirsky, B.L., Hertwich, E.G., 2018. Deriving life cycle assessment coefficients for application in integrated assessment modelling. *Environ. Model. Softw.* 99, 111–125. <https://doi.org/10.1016/j.envsoft.2017.09.010>
- Brockway, P.E., Owen, A., Brand-Correa, L.I., Hardt, L., 2019. Estimation of global final stage energy-return-on-investment for fossil fuels with comparison to renewable energy sources. *Nat. Energy* 4. <https://doi.org/10.1038/s41560-019-0425-z>
- Capellán-Pérez, I., de Castro, C., Miguel González, L.J., 2019. Dynamic Energy Return on Energy Investment (EROI) and material requirements in scenarios of global transition to renewable energies. *Energy Strateg. Rev.* 26, 100399. <https://doi.org/10.1016/j.esr.2019.100399>
- Carbajales-Dale, M., Barnhart, C.J., Brandt, A.R., Benson, S.M., 2014. A better currency for investing in a sustainable future. *Nat. Clim. Chang.* 4, 524–527. <https://doi.org/10.1038/nclimate2285>
- Castro, C. de, Capellán-Pérez, I., 2020. Standard , Point of Use , and Extended Energy Return. *Energies* 13.
- D'Alessandro, S., Cieplinski, A., Distefano, T., Dittmer, K., 2020. Feasible alternatives to green growth. *Nat. Sustain.* 3, 329–335. <https://doi.org/10.1038/s41893-020-0484-y>
- King, L.C., Van Den Bergh, J.C.J.M., 2018. Implications of net energy-return-on-investment for a low-carbon energy transition. *Nat. Energy* 3, 334–340. <https://doi.org/10.1038/s41560-018-0116-1>
- Pehl, M., Arvesen, A., Humpenöder, F., Popp, A., Hertwich, E.G., Luderer, G., 2017. Understanding future emissions from low-carbon power systems by integration of life-cycle assessment and integrated energy modelling. *Nat. Energy* 2, 939–945. <https://doi.org/10.1038/s41560-017-0032-9>
- Sgouridis, S., Carbajales-Dale, M., Csala, D., Chiesa, M., Bardi, U., 2019. Comparative net energy analysis of renewable electricity and carbon capture and storage. *Nat. Energy* 4, 456–465. <https://doi.org/10.1038/s41560-019-0365-7>

REVIEWERS' COMMENTS

Reviewer #2 (Remarks to the Author):

The authors had addressed all issues raised by me in the first revision.

After reading their subsequent response to the 1st reviewers comments, I believe that they have thoroughly addressed and explained all points raised by the 1st reviewer.

Therefore, from my perspective the authors' second revision has made all additional clarifications needed and can be published in its current form.

Response to Reviewers

Please find below our response to the comments of the one remaining reviewer. Reviewer comments are shown in *blue italics*, while our responses are in black.

Reviewer #2:

Reviewer #2 (Remarks to the Author):

The authors had addressed all issues raised by me in the first revision.

After reading their subsequent response to the 1st reviewers comments, I believe that they have thoroughly addressed and explained all points raised by the 1st reviewer.

Therefore, from my perspective the authors' second revision has made all additional clarifications needed and can be published in its current form.

Thank you for recognising our efforts in addressing your comments and those of reviewer #1. Thank you also for the suggestions in the previous round of reviews which have helped us to improve our contribution.